# IGF1-mediated human embryonic stem cell self-renewal recapitulates the embryonic niche

Sissy E. Wamaitha[1,2], Katarzyna J. Grybel[1,7], Gregorio Alanis-Lobato[1,7], Claudia Gerri[1], Sugako Ogushi [1,3], Afshan McCarthy[1], Shantha K. Mahadevaiah[3], Lyn Healy[4], Rebecca A. Lea [1], Miriam Molina-Arcas[5], Liani G. Devito[4], Kay Elder[6], Phil Snell[6], Leila Christie[6], Julian Downward [5], James M.A. Turner[3] & Kathy K. Niakan [1]*

Our understanding of the signalling pathways regulating early human development is limited, despite their fundamental biological importance. Here, we mine transcriptomics datasets to investigate signalling in the human embryo and identify expression for the insulin and insulin growth factor 1 (IGF1) receptors, along with IGF1 ligand. Consequently, we generate a minimal chemically-defined culture medium in which IGF1 together with Activin maintain self-renewal in the absence of fibroblast growth factor (FGF) signalling. Under these conditions, we derive several pluripotent stem cell lines that express pluripotency-associated genes, retain high viability and a normal karyotype, and can be genetically modified or differentiated into multiple cell lineages. We also identify active phosphoinositide 3-kinase (PI3K)/AKT/mTOR signalling in early human embryos, and in both primed and naïve pluripotent culture conditions. This demonstrates that signalling insights from human blastocysts can be used to define culture conditions that more closely recapitulate the embryonic niche.

---

[1] Human Embryo and Stem Cell Laboratory, The Francis Crick Institute, 1 Midland Road, London NW1 1AT, UK. [2] Department of Molecular, Cell and Developmental Biology, and the Eli and Edythe Broad Center of Regenerative Medicine and Stem Cell Research, University of California, Los Angeles, CA 90095, USA. [3] Sex Chromosome Biology Laboratory, The Francis Crick Institute, London NW1 1AT, UK. [4] Human Embryo and Stem Cell Unit, The Francis Crick Institute, 1 Midland Road, London NW1 1AT, UK. [5] Oncogene Biology Laboratory, The Francis Crick Institute, London NW1 1AT, UK. [6] Bourn Hall Clinic, Bourn, Cambridge CB23 2TN, UK. [7] These authors contributed equally: Katarzyna J. Grybel, Gregorio Alanis-Lobato. *email: kathy.niakan@crick.ac.uk

Understanding the signalling pathways driving early lineage specification in mouse preimplantation embryos has informed the establishment of both mouse embryonic and extraembryonic stem cell lines. For example, fibroblast growth factor (FGF) signalling and consequent mitogen-activated protein kinase (MAPK) pathway activation in the inner cell mass (ICM) regulates the divergence of pluripotent epiblast (EPI) and extra-embryonic primitive endoderm (PE) cells, that will eventually give rise to the embryo proper and yolk sac, respectively[1]. Inhibiting FGF/MAPK signalling during mouse embryo culture increases the proportion of NANOG-expressing cells within the ICM, while treatment with FGF ligands results in conversion of ICM cells to GATA6-positive PE progenitors[2,3]. In addition, leukaemia inhibitor factor (LIF) is associated with diapause, the delayed implantation of mouse preimplantation embryos[4].

Consistent with the mouse embryo, pluripotent mouse embryonic stem cells (mESCs) can be cultured in medium supplemented with LIF and inhibitors of FGF/MAPK signaling[5,6]. Addition of MEK and GSK3β inhibitors together with LIF allows mESCs to be propagated in defined medium (2i + LIF) in the absence of BMP, serum or a mouse embryonic fibroblast (MEF) supportive layer[2,7]. Conversely, culture with FGF4-supplemented medium supports the successful derivation of trophoblast stem cells from the mouse TE, and extraembryonic endoderm cells from the PE[8,9].

In contrast, only a few studies have interrogated the signalling pathways that contribute to the regulation of early human embryo cell types. TGFβ/Nodal signalling is active during early human development and has been implicated in maintaining NANOG expression in the human EPI[10], as well as regulating pluripotency gene expression in human embryonic stem cells (hESCs)[11–15]. Intriguingly, in contrast to the mouse, there does not appear to be a role for FGF signalling in the divergence of EPI and PE lineages in human embryos[16,17]. Inhibiting FGF receptors or MEK has no obvious effect on pluripotency gene expression in the human EPI[16,17]. However, FGF2 is routinely used to maintain hESCs, having initially emerged as a putative pathway of interest in experiments aiming to clonally culture hESCs[18]. The addition of FGF2 ligand is thought to be required for continuous undifferentiated proliferation in feeder-free media[19–21]. FGF is included in the majority of hESC culture medium, either by the addition of exogenous ligand, or by co-culture with mitotically inactivated fibroblast (MEF) layers, use of MEF-conditioned medium, or growth on a Matrigel matrix[22–27].

However, recently established alternative hESC culture conditions that more closely resemble those used to propagate naïve mESCs[28–34], rely on MEK inhibition together with other signalling inhibitors to maintain hESCs. While this suggests that MEK/ERK signalling may be dispensable for hESCs, these conditions rarely fully remove FGF supplementation. It is therefore unclear whether FGF signalling may still be functioning independently of MEK/ERK in this context. It has also been suggested that FGF addition only indirectly promotes hESC pluripotency, instead stimulating either the supportive MEF layer[35], or fibroblast-like cells differentiated from hESCs themselves[36], to secrete factors that subsequently promote proliferation.

We set out to investigate active signalling pathways in the human EPI, reasoning that as in the mouse, identifying these pathways in embryos would inform strategies for maintaining pluripotent hESCs in physiologically relevant conditions. Here we mine available datasets to identify receptor-ligand pairs present at the transcriptional level, in order to determine if modulating these pathways could promote either self-renewal or pluripotency. We find that IGF1 ligand and the IGF1 and insulin receptors are expressed in the human PE and EPI, respectively. IGF1 supplementation together with Activin (AI medium)

supports hESC pluripotency in chemically defined conditions without FGF addition or MEF co-culture. Under these conditions, we derive hESCs from embryos, as well as induced pluripotent stem cells (iPSCs) reprogrammed from fibroblasts. AI-cultured hESCs exhibit a transcriptional expression profile similar to hESCs cultured under conventional conditions, retain the ability to differentiate into a variety of cell types, and can be genetically modified using CRISPR/Cas9 mutagenesis. We identify active IGF signalling via phosphoinositide 3-kinase (PI3K)/mTOR in human preimplantation embryos, and in both conventional and naïve hESC culture conditions, and determine that inhibiting FGF receptors does not adversely affect hESC maintenance. The successful generation of these pluripotent self-renewing cell lines demonstrates the utility of our approach of integrating signalling insights from the human early embryo to design minimal FGF-free hESC culture conditions that better reflect the pluripotent niche.

## Results

**IGF signalling components are expressed in the human epiblast.** We interrogated published single-cell early human transcriptome datasets[10,37] to identify lineage-associated gene expression patterns in the human blastocyst, comparing the human pluripotent EPI to the extraembryonic TE and the PE. We previously showed that several TGFβ/NODAL signalling-associated genes were enriched in the EPI, and that intact TGFβ/Nodal signalling is required for pluripotency gene expression in the human embryo[10], as had been described in hESCs[13]. We therefore reasoned that our transcriptome datasets could identify signalling pathways in the human embryo that might have relevance for the establishment or maintenance of hESCs.

To identify EPI, PE or TE-associated signalling networks, we performed a comparative gene set enrichment analysis (GSEA)[38] and visualised pathway relationships using the EnrichmentMap tool[39] in Cytoscape[40] (Supplementary Fig. 1, Supplementary Data 1). We overlaid gene expression, represented as boxplots, onto KEGG pathway maps[41] thereby creating a searchable visual representation of expression for each pathway component in each cell lineage (https://shiny.crick.ac.uk/embryo_signalling/). To identify ligand and cognate receptor pairs that might suggest signalling pathways to modulate, we curated a list of ligands and receptors from all signal transduction pathways in the KEGG database, and integrated protein-protein interactions using the HIPPIE database[42] (Supplementary Data 2). This represents a comprehensive analysis of signalling pathway-associated transcripts that are expressed in human preimplantation embryos at a time when three distinct cell types are becoming specified in their fate and function.

As in our previous study[10], our analysis indicated that pathways associated with TGFβ/NODAL signalling were enriched in the EPI compared to the TE. Furthermore, pathways related to regulation of IGF activity, JAK/STAT signalling and interleukin signalling were enriched in the EPI and PE compared to the TE (Supplementary Data 1). Transcripts for the insulin (INSR) and IGF1 (IGF1R) receptors were more highly enriched in the EPI, while insulin-like growth factor 1 (IGF1) ligand was enriched in the PE (Fig. 1a). The interleukin 6 receptor subunit (IL6R), which can colocalise with IGF receptors to induce Akt-mediated proliferation in alternative contexts[43,44], was also enriched in the EPI (Supplementary Data 1). IL-6 activity has been shown to regulate downstream JAK/STATs[45] and to crosstalk with insulin/IGF1[43]. The type II diabetes pathway was also identified as enriched in the EPI; this includes genes encoding insulin and IGF receptors as well as downstream PI3K signalling effectors

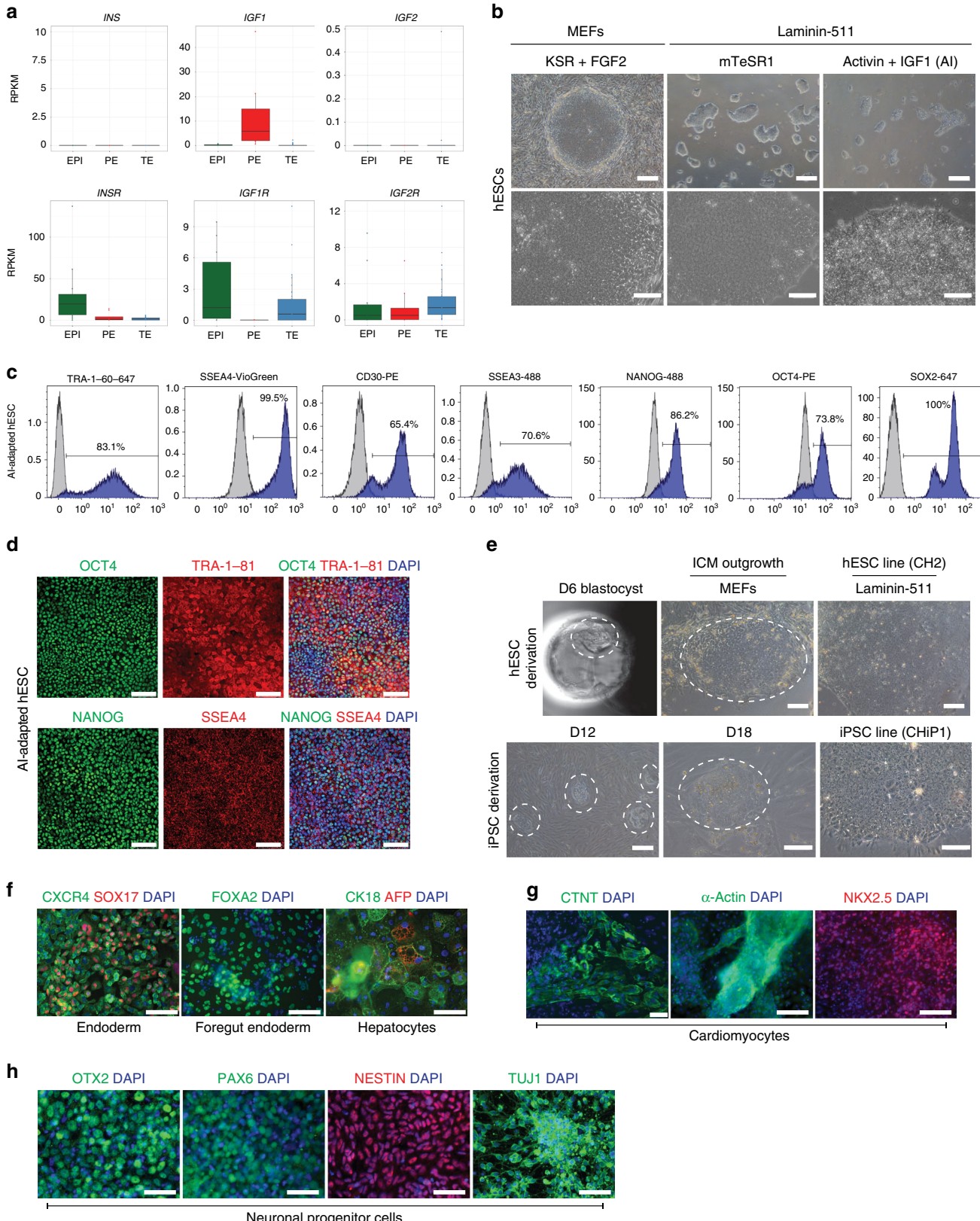

(Supplementary Data 1), and further suggested that modulating IGF signalling would be of interest.

Enrichment analysis showed that genes related to negative regulation of FGFR signalling were present in the EPI, which could reflect negative feedback loops indicating pathway activation (Supplementary Data 1). However, we did not find FGF-related receptors specifically highly enriched in the EPI. Boxplot analysis showed *FGFR1*, *FGFR2* and *FGFR4* were more highly expressed in the PE, while transcripts for FGF ligands, with the exception of *FGF18*, were not expressed above the required 5 RPKM threshold (Supplementary Fig. 3). The expression of downstream MAPK pathway components (e.g. *KRAS* and

**Fig. 1 A chemically defined hESC medium comprising Activin and IGF. a** Boxplots of RPKM values for insulin (*INS*) and IGF ligands (*IGF1, IGF2*), and cognate receptors in human blastocyst lineages (EPI green, PE red, TE blue) determined by single-cell RNA-seq[10]. Boxes symbolise first and third quartiles, horizontal lines the median, whiskers extend to 1.5 times the interquartile range, dots represent outliers. **b** Representative phase-contrast images of hESCs in KSR + FGF2 on MEFs, or mTeSR1 or AI medium on laminin-511. *n* = 2 or 3 biological and 3 technical replicates. Scale bar: 300 μm, top row; 200 μm bottom row. **c** Flow cytometry of AI-adapted Shef6 hESCs for the indicated proteins (blue); isotype control in grey. *n* = 3 technical replicates. **d** Representative immunofluorescence of AI-adapted Shef6 hESCs for the indicated proteins; DAPI (blue) nuclear stain. *n* = 3 biological replicates. Scale bar: 100 μm. **e** Upper panels: Representative image of a human day 6 blastocyst prior to ICM (circled) dissection and plating in AI medium on MEFs. The resulting ICM outgrowth (circled) was passaged onto laminin-511 to establish a stable hESC cell line (CH2). Scale bar: 150 μm. *n* = 3 biological replicates. Lower panels: Representative images of iPSCs derived following reprogramming of BJ fibroblasts with Sendai viruses. iPSC-like colonies (circled) could be detected in AI medium within 12 days of induction, and expanded for picking within 18 days to propagate an established iPSC line (CHiP1). *n* = 2 technical replicates. Scale bars, left to right: 300 μm, 100 μm, 100 μm. **f–h** Representative immunofluorescence of AI-cultured hESCs following directed differentiation. *n* = 3 biological replicates. DAPI in blue. **f** hESCs differentiated towards hepatocyte-like cells first express endoderm markers CXCR4 (green) and SOX17 (red), upregulate FOXA2 (green) as foregut endoderm, then AFP (red) and CK18 (green) as hepatocytes. Scale bar: 100 μm. **g** Expression of CTNT (green; scale bar 50 μm), α-Actin (green; scale bar 100 μm) and NKX2.5 (red; scale bar 100 μm) following directed differentiation towards cardiomyocyte-like cells. **h** Expression of neuronal markers OTX2, PAX6, TUJ1 (green) and NESTIN (red) following directed differentiation towards neuronal progenitor cells. Scale bar: 100 μm.

*MAPK1*) suggests that Receptor Tyrosine Kinase (RTK) signalling may be activated by alternative receptors or via signalling pathway cross-talk.

Given the enrichment of IGF/insulin signalling-related components, we set out to develop a minimal hESC culture medium using TGFβ/Nodal, with IGF in place of FGF. As the expression of insulin, IGF2 and IGF2R transcripts were either undetectable or not specifically expressed in EPI cells (Fig. 1a), we chose to proceed using IGF1 ligand.

**A chemically defined minimal hESC medium comprising Activin and IGF1.** We first sought to select an appropriate basement membrane substrate for hESC maintenance, interrogating the transcriptome dataset to identify enriched known integrin binding partners of commonly used basement membrane proteins[46,47]. Transcripts for laminin-binding Integrin-α6β1 (*ITGA6, ITGB1*) were expressed in the human EPI, as well as the TE and PE (Supplementary Fig. 4a). However, integrin subunits integrin-α5 (*ITGA5*) and integrin-αV (*ITGAV*), which complex with ITGB1 as Integrin-α5β1 and Integrin-αVβ1 to bind fibronectin and vitronectin respectively, were not detected in the EPI (Supplementary Fig. 4a). This suggests that pluripotent cells in vivo may be specifically receptive to laminins. We adapted hESCs to either whole recombinant human laminin-511[48] or a recombinant laminin-511 E8 fragment[49,50], and found that, as in previous studies, these maintained hESC morphology comparable to those on Matrigel (Supplementary Fig. 4b). We therefore continued with this embryo-specific basement membrane component for hESC culture in chemically defined conditions.

We next sought to establish a viable culture system using IGF1 and Activin. hESCs were grown on laminin-511 in either control mTeSR1 medium, or adapted to basal medium alone (Advanced-DMEM/F12 plus 2 mM glutamine supplement); basal medium plus 10 ng/ml Activin and 12 ng/ml FGF2 (previously published growth factor concentrations[12]); basal medium plus 17 nM IGF1 (concentration based on calculations from human uterine fluid[51]) or basal medium plus 10 ng/ml Activin and 17 nM IGF1 (Supplementary Fig. 4c). hESCs cultured in basal medium plus Activin and IGF1 (hereafter AI medium) resembled control cells in mTeSR1 or Knockout serum replacement (KSR) and FGF2 (KSR + FGF2) media, with tightly packed colonies and distinct colony boundaries (Fig. 1b, Supplementary Fig. 4c). hESCs in AI medium could be propagated by passaging in clumps via manual picking or using Ca+/Mg+-free PBS, Gentle Cell Dissociation Buffer or ReLeSR reagents, but not EDTA (Supplementary Fig. 4d). Single cells could also be disaggregated with Accumax in the

presence of a Rho-associated kinase (ROCK) inhibitor, and then further propagated equivalent to those passaged as clumps (Supplementary Fig. 4e).

Independent hESCs (H1, H9, Shef6) and iPSCs (RCiB10) adapted to AI medium have been propagated for multiple passages, including lines that have been maintained for over 24 months and over 70 passages. Cells were passaged on an approximate 4 to 5-day cycle using Accumax single-cell dissociation with ROCK inhibitor and retained a high nuclear-to-cytoplasmic ratio (Supplementary Fig. 4e). Immunofluorescence and flow cytometry analysis determined that hESCs and iPSCs adapted to AI medium robustly expressed pluripotency-associated factors NANOG, OCT4 and SOX2, and the cell surface antigens TRA-1-60, SSEA4, CD30 and SSEA3, in proportions comparable to hESCs cultured in mTeSR1 or KSR + FGF2 (Fig. 1c, d, Supplementary Fig. 5a, b). This was maintained whether cells were passaged as single cells or in clumps, suggesting AI medium could also support the establishment of clonal cell lines and the derivation of transgenic or genetically modified hESCs. To test this, we nucleofected hESCs cultured in AI medium with a plasmid engineered to express the Cas9 gene and a guide RNA targeting a non-essential gene in hESCs (*ARGFX*) plus a puromycin selection cassette. After 48 h of puromycin selection and subsequent clonal expansion for 10 days, colonies were confirmed to be positive for alkaline phosphatase (Supplementary Fig. 5c). A T7 endonuclease assay confirmed that the hESCs cultured in AI medium had been successfully targeted (Supplementary Fig. 5d).

G-band karyotype analysis and whole-genome sequencing confirmed that both hESCs and iPSCs cultured in AI medium for multiple passages retained a normal complement of 46 chromosomes (Supplementary Fig. 5e–g). AI-cultured hESCs and iPSCs retained high viability and proliferated similar to cells cultured in mTeSR1 or KSR + FGF2 (Supplementary Fig. 5h). Altogether, these data illustrate that a minimal medium comprising Activin and IGF1 (AI) is sufficient to support pluripotent cell cultures.

**AI medium supports de novo hESC-derivation and iPSC reprogramming.** We next asked whether AI medium could be used to derive hESCs from human embryos. Blastocysts were cultured overnight in human embryo culture medium. The ICM and overlaying polar TE were then microdissected from the mural TE, and plated in AI medium on MEFs (Fig. 1e). hESC-like colonies emerged within a week of plating and were maintained thereafter as stable hESC lines on laminin-511 (Fig. 1e). Immunofluorescence and flow cytometry analysis confirmed that AI-derived hESCs expressed pluripotency-associated intracellular

and cell surface markers, and were comparable to H9 cells in mTeSR1, KSR + FGF2 or naïve t2iL + Gö conditions (Supplementary Figs. 5i, 6). Three stable lines, CH1, CH2 and CH3 were derived under AI conditions.

We also evaluated whether AI medium could support derivation of iPSCs from human fibroblasts using two standard reprogramming methods. In the first instance, we virally transduced BJ new-born foreskin fibroblasts (ATCC®CRL-2522) with Sendai virus vectors harbouring the reprogramming factors OCT4, SOX2, KLF4 and c-MYC[52]. Following transduction, fibroblast cells were re-plated into KSR + FGF2, TeSR-E8 or AI medium. Colonies resembling pluripotent stem cells emerged within 12 days (Fig. 1e, Supplementary Fig. 5j, circled) and expressed TRA-1-60, confirming their iPSC-like identity (Supplementary Fig. 5k). Colonies in AI medium were confirmed to be positive for alkaline phosphatase expression (Supplementary Fig. 5l), continued to expand over time and were passaged with Accutase to establish stable iPSC lines (Fig. 1e).

We also reprogrammed a second foetal lung fibroblast cell line MRC5 (ECACC 05072101) with non-modified RNAs[53], using either AI or Nutristem® medium. Morphological changes were observed from day 3 of the reprogramming timeline, and by day 9, iPSC colonies were evident under both conditions. Colonies were shown to be positive for alkaline phosphatase activity on day 34 (Supplementary Fig. 5m), and were further propagated on laminin-511 and passaged using TrypLE™ Express (Supplementary Fig. 5n). Altogether, this demonstrates that AI medium supports the establishment of both self-renewing iPSC colonies and hESCs from human embryos.

**AI hESCs respond effectively to differentiation cues.** We sought to determine whether AI-adapted hESCs remained capable of responding to directed differentiation cues. We first evaluated their propensity to differentiate into hepatocyte-like cells, using a standard protocol[54]. Immunofluorescence analysis confirmed emergence of initial pan-endoderm cells expressing SOX17 and CXCR4, followed by foregut endoderm cells expressing FOXA2, and eventually hepatocyte-like cells expressing alpha-fetoprotein (AFP) and cytokeratin-18 (CK18) (Fig. 1f).

We also differentiated AI-adapted hESCs into cardiomyocytes using the STEMdiff cardiomyocyte differentiation kit. qRT-PCR analysis confirmed upregulation of GATA4, ISL1, TBX5, MEF2C, MLC2A and TNNT2 lineage markers, and downregulation of NANOG, OCT4 and SOX2 as differentiation progressed (Supplementary Fig. 7a). Differentiated cells displayed typical cardiomyocyte morphology, growing as an adherent tightly packed monolayer that contracted in culture (Supplementary Movie 1), indicating they had acquired the potential for electrical activity. Immunofluorescence analysis confirmed the expression of cardiac muscle markers NKX2.5, α-Actin and cardiac troponin (CTNT) (Fig. 1g).

Finally, we used a standard dual SMAD inhibition protocol to generate neuronal progenitor cells from AI-adapted hESCs[55]. We detected the expression of OTX2, PAX6, NESTIN and TUJ1 by immunofluorescence, confirming the neuronal identity of the differentiated cells (Fig. 1h).

In addition, AI-derived hESCs were allowed to spontaneously differentiate by culturing in MEF medium for up to 2 weeks. Immunofluorescence analysis confirmed the emergence of SOX17-expressing endoderm cells, TUJ1-expressing ectodermally-derived neurons and DESMIN-expressing mesoderm cells (Supplementary Fig. 7b). Altogether, we confirmed that AI-cultured hESCs retained the capacity to differentiate into multiple cell lineages.

**AI hESCs are transcriptionally similar to conventional hESCs.** We next compared the transcriptome of individual AI-cultured hESCs to published single-cell datasets from human blastocyst embryos[10,37,56]. As a comparison, we also included hESCs cultured in conventional KSR + FGF2 on MEFs or mTeSR1 media on either Matrigel or laminin-511.

After adjusting for batch effects in the expression data, we used 3087 variably expressed genes to perform dimensionality reduction analyses. Principal component analysis (PCA) indicated that principal component 1 (PC1) separated hESCs and blastocyst samples, while PC2 and PC3 distinguished hESCS in AI and mTeSR1 from hESCs cultured in KSR + FGF2 media (Supplementary Fig. 8a). These patterns were confirmed using uniform manifold approximation and projection[57] (UMAP), a non-linear dimensionality reduction method (Supplementary Fig. 8b). Controlling for confounding sources of variance (e.g. cell cycle) using the graph inference of population heterogeneity (griph) clustering tool indicated that AI-cultured hESCs were transcriptionally similar to mTeSR1-cultured hESCs, somewhat distinct from cells in KSR + FGF2, and comparatively distinct from blastocyst cells (Supplementary Fig. 8c).

We next compared global gene expression of AI-cultured hESCs to hESCs in naïve hESC culture medium by integrating several published datasets[28,33,58]. The griph clustering tool indicated that the first dimension (Dim1) separated all hESCs from the embryo EPI, PE and TE cells (Fig. 2a). In the second dimension (Dim2), naïve hESCs clustered distinctly compared to hESCs cultured in AI, mTeSR1 or KSR + FGF media. We also included a comparison to a cynomolgus monkey embryo dataset[59], where it was suggested that hESCs in conventional conditions cluster more closely to the post-implantation cynomolgus monkey EPI, while those in naïve conditions are more similar to the pre-implantation compartment. PCA and UMAP analysis indicated that hESCs in AI and mTeSR1 clustered together, while cynomolgus monkey ESCs and hESCs in naïve conditions clustered with the post-implantation cynomolgus EPI (Supplementary Fig. 8d). Both these groups were distinct from the human EPI and cynomolgus pre-implantation EPI samples.

To further characterise each cell type, we performed pair-wise differential gene expression analysis using DESeq2 to compare hESCs cultured in AI medium, mTeSR1 on Matrigel or laminin-511, KSR + FGF2 or naïve hESC medium, with the embryonic EPI, TE and PE cells (Supplementary Data 3). The genes most differentially expressed were used to perform a GSEA using a compendium of multiple signalling and pathway databases[40] (Supplementary Data 4), as well as functional enrichment analysis using Gene Ontology (GO)[60] and REACTOME[61] databases (Fig. 2b, Supplementary Fig. 9a). Genes associated with extra-cellular matrix organisation were enriched in AI-cultured hESCs, consistent with the use of laminin basement membrane components for adherent culture (Fig. 2b). Interestingly, genes associated with the WNT signalling pathway were more highly enriched in either AI- or mTeSR1-cultured hESCs on laminin compared to the EPI (Supplementary Figs. 2b, 9b). Genes associated with cholesterol biosynthetic processes were enriched in mTeSR1-cultured hESCs on Matrigel compared to the EPI (Supplementary Fig. 9c). In contrast to AI-cultured hESCs, the EPI was enriched for genes associated with RNA binding and transport, histone binding, translation initiation complex formation and activation of mRNA, suggesting differences in transcription and translation. The EPI was also enriched for genes associated with antiviral mechanisms, interferon-stimulated genes and the immune system (Supplementary Data 4; Supplementary Fig. 9a). Consistent with the PCA, UMAP and griph analyses, mTeSR1-cultured hESCs on Matrigel or laminin-511 were transcriptionally similar to each other and differentially

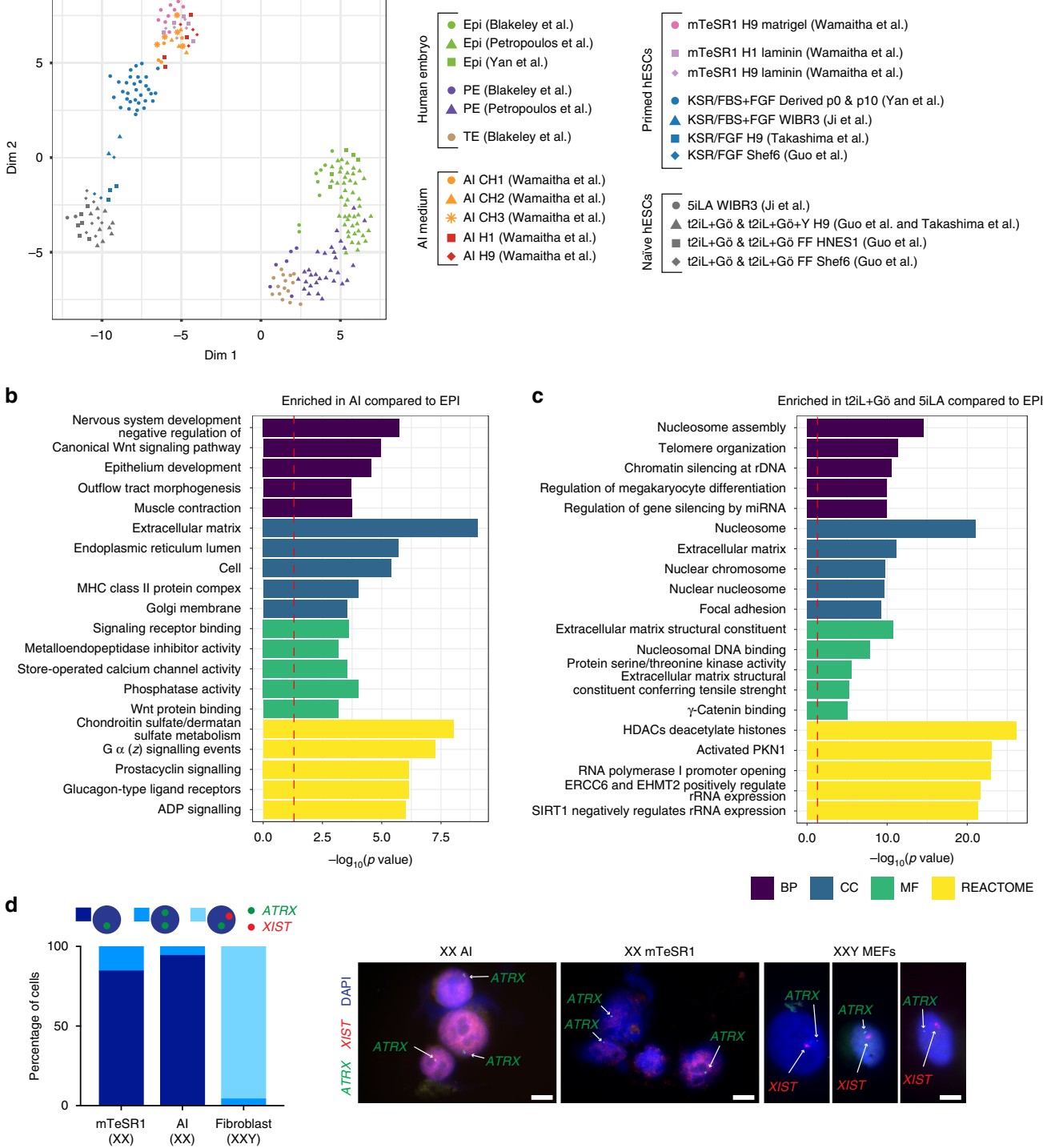

**Fig. 2 hESCs cultured in AI medium are transcriptionally similar to conventional hESCs. a** Clusters detected after applying the unsupervised clustering tool griph to single-cell RNA-seq data from EPI, PE or TE cells of the human blastocyst[10,37,56] and hESCs cultured in AI, mTeSR1, KSR + FGF, 5iLA or t2iL + Gö media[28,33,37,58]. A colour-coded sample key is provided. **b, c** Comparison of embryonic EPI cells to cells cultured in AI medium or naïve (5iLA or t2iL + Gö) media. The most enriched GO terms for genes associated with biological process (BP), cellular component (CC), molecular function (MF), and REACTOME pathways associated with differentially expressed genes are shown based on Benjamini-Hochberg corrected p-values. Red-dashed lines correspond to the significance level $\alpha = 0.05$. **d** Quantification of RNA FISH in XXY fibroblast cells or in H9 XX hESC lines cultured in mTeSR1 or AI media for the transcription foci of *XIST* and *ATRX*, which is normally subject to X-chromosome inactivation. $n = 51$ mTeSR1 hESCs, 50 AI hESCs and 51 fibroblasts. Representative DAPI and RNA FISH images of hESCs or fibroblasts are shown. Scale bar: 10 μm. Source data are provided as a Source Data file.

expressed genes were mostly associated with mitochondrial function (Supplementary Data 5).

Functional enrichment analysis also highlighted that genes associated with DNA methylation, histone deacetylases (HDACs), RNA polymerase I promoter opening and nucleosome DNA binding and assembly were more highly enriched in naïve hESCs compared to the human EPI (Fig. 2c). This is consistent with the distinct epigenetic state and global hypomethylation of naïve hESCs[33,62]. By contrast, the genes more highly expressed in human EPI cells were associated with mitochondrial function and biogenesis as well as RNA polymerase activity and mRNA splicing, suggesting differences in metabolism and transcriptional regulation, compared to in vitro cultured naïve hESCs (Supplementary Fig. 9d).

Consistent with previous publications[63], we confirm that all hESCs are transcriptionally distinct from EPI cells, irrespective of the culture conditions, suggesting that further refinement of hESC medium is needed to capture a more similar pluripotent state. Altogether, the clustering analysis suggests that AI-cultured hESCs are transcriptionally more similar to hESCs cultured in conventional mTeSR1 or KSR + FGF medium, than to hESCs cultured in naïve conditions. This suggests that IGF1 and FGF2 may activate overlapping downstream mechanisms for the maintenance of hESCs.

**AI- or mTeSR1-cultured hESCs exhibit a similar X-chromosome inactivation state**. hESCs cultured in conventional versus naïve culture conditions exhibit distinct X-inactivation states[64]. Conventional primed female hESCs typically carry one active and one inactive X-chromosome and the X-inactivation regulator XIST-RNA coating is often absent[65], while naïve cells reactivate the inactive X-chromosome and exhibit mono-allelic XIST expression[64]. To characterise X-chromosome activity in AI-cultured hESCs, we performed dual RNA fluorescence in situ hybridization (RNA-FISH) for XIST and the X-linked gene ATRX. We compared our findings with those observed in mTeSR1-cultured XX hESCs, and in XXY fibroblasts (Fig. 2d). We observed mono-allelic expression of ATRX and XIST in the majority of XXY fibroblasts, consistent with the existence of X-inactivation. In the majority of AI-cultured hESCs, ATRX was expressed mono-allelically, and XIST clouds were absent. Similar findings were made in mTeSR1 hESCs, consistent with previous findings[65]. The X-inactivation status of AI-cultured hESCs therefore resembles that of hESCs cultured in more conventional conditions.

**PI3K/mTOR signalling downstream of IGF1 regulates hESC self-renewal**. We next investigated mechanisms by which IGF1 supports hESC self-renewal. Canonically, binding of insulin or IGF ligands to their receptors results in phosphorylation and activation of PI3K[66] (Fig. 3a). PI3K phosphorylates the membrane-bound phospholipid PIP2 to generate active PIP3, which then activates a kinase cascade via phosphoinositide-dependent kinase-1 (PDK1) and its substrates AKT and atypical protein kinase Cs (aPKCs). Downstream AKT substrates include mTOR, FOXO1, S6 and GSK3β[66], and mTOR signalling has been implicated in promoting hESC self-renewal and inhibiting apoptosis[67,68]. Notably, these previous studies were performed in hESC medium containing exogenous FGF, which can also activate PI3K downstream of FGF receptor binding[69]. Considering the similarities between cells cultured in AI and mTeSR1 revealed in our transcriptional analyses, we reasoned they might also exhibit common signalling dynamics.

As expected, in AI-cultured hESCs we observed phosphorylation of the insulin and IGF1 receptors (IR and IGF1R), as well as the downstream pathway components AKT-T308, AKT-S473

(pAKT) and S6 (pS6), confirming PI3K signalling pathway activity (Fig. 3b). We also detected phosphorylated ERK1/2 (pERK), indicating active MAPK signalling (Fig. 3b), possibly as a consequence of crosstalk between the PI3K and MAPK signalling pathways. Both the IGF1R/insulin/PI3K and MAPK pathways were also active in mTeSR1-cultured hESCs (Supplementary Fig. 10a).

To determine whether AI-cultured cells were responsive to the modulation of these pathways, we employed small molecule inhibitors targeting various pathway components (Fig. 3b). IGF1R/IR phosphorylation was reduced within 24 h of treatment with the IGF1R/IR inhibitor OSI-906 (OSI), as was downstream pAKT. Increased pERK was observed following OSI treatment, suggesting crosstalk between the PI3K and MAPK signalling pathways. Reduced pERK levels were observed within 24 h of treatment with the FGF receptor tyrosine kinase inhibitor PD173074 (PD17) or MEK inhibitor PD0325901 (PD03) (Fig. 3b). Inhibition of mTOR with Everolimus led to the loss of downstream pS6, as well as increased pAKT and pERK, suggesting potential feedback to both of these pathways (Fig. 3b). mTeSR1-cultured hESCs were similarly responsive to modulation of these signalling pathway components (Supplementary Fig. 10a).

We next treated AI- or mTeSR1-cultured hESCs with OSI, Everolimus, PD17, or PD03 for 3 days to assess the impact on proliferation over this time. Treatment with OSI or Everolimus impacted both the morphology and proliferation of AI- or mTeSR1-cultured hESCs (Fig. 3c, Supplementary Fig. 10b). PD03 treatment impacted proliferation in both culture media, suggesting that MEK/ERK signalling is required for hESC self-renewal in these conditions (Fig. 3c). However, treatment with PD17 had no discernible effect on proliferation in either mTeSR1 or AI medium (Fig. 3c) indicating that hESCs can be maintained independent of FGF ligand availability. Although we observed some downregulation of pERK following PD17 treatment, suggesting FGFR kinase activity does feed into pERK in AI conditions, this does not seem to be required for cell survival and proliferation. The absence of any obvious loss of NANOG under these inhibitor conditions (Fig. 3d) points to the specific role of IGF1/insulin in regulating hESC proliferation.

We next sought to investigate the expression of these pathways in naïve culture conditions, using hESCs in t2iL + Gö medium (Supplementary Fig. 10c). pIGF1R/pIR, pAKT and pS6 were all detectable in hESCs cultured in t2iL + Gö on Geltrex (Supplementary Fig. 10d). Modulation with OSI, GDC or Everolimus resulted in the inhibition of pIGFR1/pIR, pAKT or pS6 respectively, indicating that naïve cells are also responsive to PI3K signalling modulation. Surprisingly, we detected pERK expression in hESCs cultured in t2iL + Gö, despite the presence of PD03 in these conditions (Supplementary Fig. 10d), suggesting the concentration of PD03 used does not fully block MEK/ERK signalling. Consistent with this, treatment with PD17 led to downregulation of pERK, suggesting that MEK/ERK signalling may still be active. Treatment of naïve cells with inhibitors for 3 days had no obvious impact on cell proliferation or NANOG expression (Fig. 3c, d), although as long-term treatment has yet to be assessed, there remains the possibility that prolonged inhibitor treatment may impact self-renewal. Altogether our results indicate that PI3K/mTOR signalling is active in both conventional and naïve hESC culture conditions.

**IGF1 regulates the proliferation of the human ICM**. Given the positive effect of IGF signalling on hESC proliferation and our identification of PI3K/mTOR activity across hESC culture conditions, we returned to the human embryo to interrogate the role

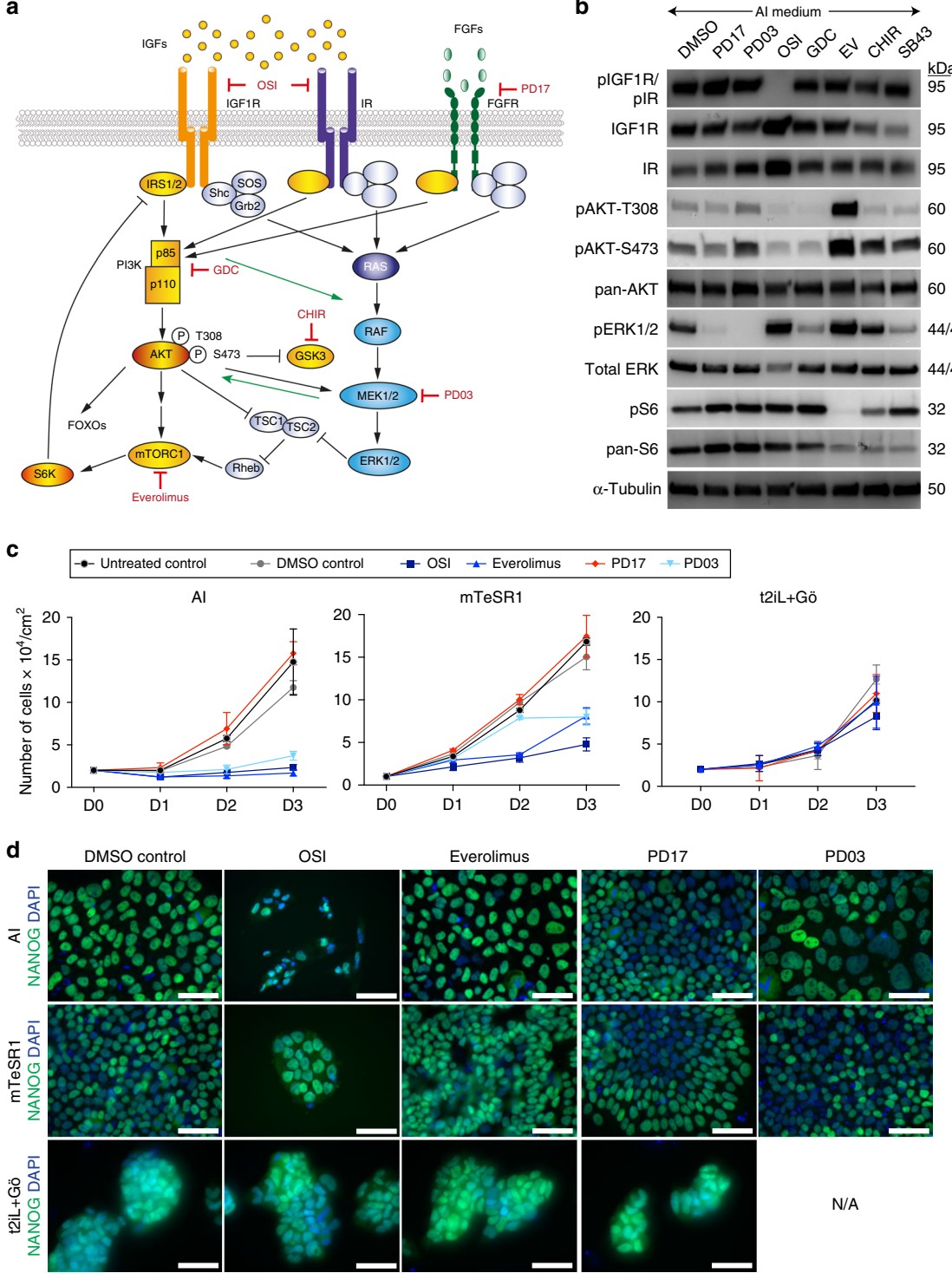

**Fig. 3 Insulin/IGF1R/PI3K/mTOR and MAPK/ERK signalling pathways are active in both conventional and naïve hESC culture conditions. a** Schematic of the crosstalk between the insulin/IGF1 and receptor tyrosine kinase (RTK) signalling pathways. Insulin receptor (IR), IGF1 receptor (IFG1R), IGF binding proteins (IGFBPs), receptor tyrosine kinase (RTK), IGFR and IR inhibitor OSI-906 (OSI), FGF receptor tyrosine kinase inhibitor PD173074 (PD17), PI3K inhibitor GDC-0941 (GDC), GSK3β inhibitor CHIR99021 (CHIR), MEK inhibitor PD0325901 (PD13), mTOR1 kinase inhibitor Everolimus (EV). **b** Representative western blot analysis for proteins related to PI3K/AKT/mTOR and MAPK/ERK signalling in hESCs cultured in AI medium supplemented with DMSO, PD17, PD03, OSI, GDC, EV, CHIR or the Activin/Nodal receptor inhibitor SB-431542 (SB43) for 24 h. $n = 3$ biological replicates. **c** Quantification of the total number of viable cells over 3 days in AI, mTeSR1 or t2iL + Gö media supplemented with DMSO, OSI, EV, PD17 or PD03. The value at each time point represents the mean of $n = 2$ biological and $n = 4$ technical replicates, and error bars represent s.e.m. **d** Representative immunofluorescence analysis for NANOG (green) expression in control or inhibitor-treated hESCs in AI, mTeSR1 or t2iL + Gö media, following 3 days of treatment. DAPI (blue) nuclear stain. Scale bar: 50 μm. Source data are provided as a Source Data file.

of IGF1. Insulin/IGF1 signalling has been implicated in embryo development, with studies suggesting that IGF1 may be present in the fallopian tube and uterine fluid[51,70]. Studies treating human embryos with IGF1 identified an increase in ICM cell numbers as well as an increase in the proportion of embryos developing to the blastocyst stage[70,71]. However, these studies did not investigate pluripotency gene expression under these conditions.

We first investigated PI3K/mTOR signalling activity in human embryos, developing a method based on a refined western blot technique. Preliminary tests on mouse embryos were able to reproducibly detect pAKT and pS6 expression from low numbers of blastocysts (Fig. 4a). Human blastocysts were then collected at 7 days post fertilisation (dpf) and assayed, and we detected phosphorylated AKT and S6 (Fig. 4a). Coupled with our initial transcriptomics data showing IGF1 and insulin receptor enrichment, this confirmed that IGF/PI3K signalling was likely to function in human embryos. Given the cross-talk we observed between downstream AKT/mTOR and MEK/ERK, we next assayed pERK in mouse and human embryos and confirmed its expression (Fig. 4a), suggesting activation of these pathways.

We next cultured four- to eight-cell human embryos at 2 dpf in human embryo culture medium supplemented with 17 nM IGF1 (Fig. 4b). By 6.5 dpf, treated embryos had formed an expanded blastocyst and expressed NANOG in the EPI, and the PE marker SOX17, similar to controls (Fig. 4c). Embryos also expressed KLF17 (Fig. 4d), a transcription factor we previously identified as enriched in the EPI[10]. The automated software tool MINS 1.3[72] was used to detect and segment DAPI-stained nuclei to determine embryo cell number. The ICM was identified by morphology, and ICM cell number determined by tallying NANOG- or SOX17-expressing cells. By 6.5 dpf the ICM proportion was increased in IGF1-treated embryos compared to controls (Fig. 4e, $P = 0.034$, one-tailed $t$-test). IGF1 treatment also resulted in a 2-fold increase ($P = 0.024$) in the proportion of NANOG-expressing cells (Fig. 4e). The proportions of SOX17-expressing cells were not similarly affected ($P = 0.083$) and total embryo cell number was equivalent to controls (Fig. 4e). In summary, IGF1 treatment has a demonstrable proliferative effect in the ICM of the human embryo.

As a comparison, we treated human embryos with exogenous FGF. Although inhibiting FGF/ERK signalling in human blastocysts does not impact pluripotency gene expression[16,17], FGF could perhaps promote ICM proliferation. Previous experiments in the mouse showed that addition of 1 μg/ml FGF2 and 1 μg/ml heparin to embryo culture media resulted in a cell-fate conversion of NANOG-expressing EPI cells to GATA6-expressing PE cells by the blastocyst stage[3]. Therefore, in the first instance a similar treatment schedule was applied to 4- to 8-cell human embryos (2 dpf).

By 6.5 dpf, untreated control embryos formed an expanded blastocyst and expressed both NANOG and GATA6 (Fig. 4f). In contrast, FGF-treated embryos did not express NANOG and were smaller compared to controls. This differs from FGF-treated mouse embryos, which appear developmentally normal despite lineage conversion within the ICM[3]. Embryos were next treated with 100 ng/ml FGF2, a concentration commonly used to maintain hESC pluripotency in vitro[27]. Again, by 6.5 dpf treated embryos were smaller and lacked NANOG-expressing cells but retained GATA6-expressing cells (Fig. 4f), indicating a negative effect on pluripotency gene expression.

To determine if FGF stimulation affected established NANOG expression, embryos were treated with 100 ng/ml FGF2 from 5 dpf, by which time NANOG expression is restricted to the ICM[73]. Treated embryos appeared developmentally normal and NANOG expression was maintained in the ICM (Fig. 4g). Analysis using MINS 1.3[72] determined that proportions of both NANOG- and

SOX17-expressing cells and the total ICM were within the range of controls (Fig. 4h; NANOG $P = 0.150$, SOX17 $P = 0.348$, ICM $P = 0.150$). Altogether, this suggests that rather than operating via an alternative pathway in the human embryo, FGF is unlikely to be crucial for EPI development.

## Discussion

The initial investigations that established conditions for hESC derivation were carried out in non-human primates[74], and these have been transformative in enabling the establishment of hundreds of hESC lines worldwide[22]. However, these conditions may not necessarily reflect the requirements of early human embryonic development. Several new culture systems have recently been identified primarily via chemical screening methods using existing stem cell lines[28–34]. We took an alternative approach by analysing gene expression patterns and modulating signalling in the human EPI directly, which we hypothesised would help clarify the role of signalling in the embryo and elucidate which pathways are recapitulated in in vitro derived hESC lines. The signalling datasets we generated may inform further refinement of hESC culture conditions, as well as optimisation of human trophoblast[75] and extraembryonic endoderm[76] stem cell media.

We show that IGF1 supplementation coupled with Activin supports derivation and maintenance of human pluripotent stem cells without the addition of exogenous FGF2. Interestingly, despite displaying distinct morphology from mTeSR1-cultured hESCs, AI hESCs were transcriptionally similar. This suggests that scoring colonies based on morphology may not always accurately reflect their transcriptional state, or self-renewal and differentiation potential, consistent with previous findings[77]. AI medium could also represent a simple yet refined alternative that could be produced in any lab, in comparison to cost-prohibitive commercial media. Importantly, the viability or proliferative capacity of AI- or mTeSR1-cultured hESCs was unaffected by FGF receptor inhibitors, in stark contrast to their requirement for IGF1/insulin signalling, indicating IGF1/insulin signalling via AKT/mTOR is necessary for hESC maintenance. Inhibiting MEK impedes self-renewal in both AI and mTeSR1, indicating that a mechanism involving this kinase plays a key role. However, it is unclear whether ERK is involved, as pERK is also downregulated following FGFR inhibition, where there is no effect on hESC viability. Although ERK is commonly the master effector of the MEK pathway, alternative substrates have been recently identified, such as heat-shock transcription factor 1 (HSF1)[78]. Furthermore, while a few studies have implicated downstream mTOR signalling components and FOXO proteins in the regulation of hESC viability and self-renewal[68,79], additional study will elucidate whether other downstream effectors may also be required.

Interestingly, mTeSR1 medium contains insulin[27], as do several other hESC media, supplements and basement membrane substrates[27,80–84], suggesting this pathway is being stimulated to support hESC self-renewal in existing culture conditions. Previous studies showed that inhibition of active PI3K/AKT signalling in hESCs is required for differentiation[85]. IGF1 has also been shown to confer an additional benefit in standard hESC medium that contains exogenous FGF[68,85,86]. Adding an IGF1R-blocking antibody to hESC cultures impeded the expansion of cells expressing the pluripotency-associated marker SSEA3, which could not be rescued by exogenous FGF2 supplementation[36]. A combination of IGF1 and heregulin has also been shown to reproduce the same positive effect on long-term hESC self-renewal as high levels of FGF2[87]. Notably, our conditions lack heregulin, which is a member of the epidermal growth factor

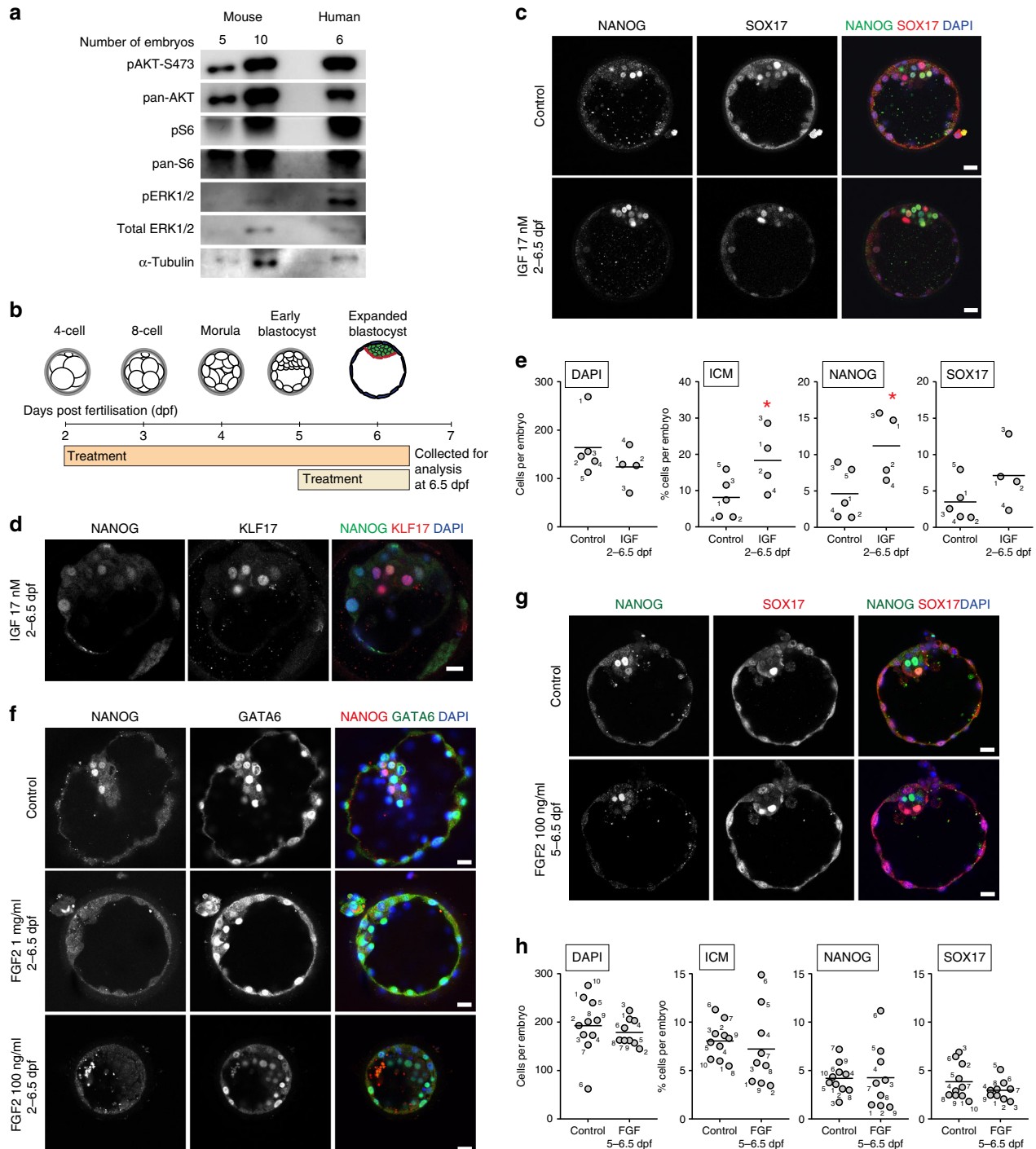

(EGF) family and thereby can activate the RTKs, similarly to FGFs. While these studies suggested a cooperative role of FGF or heregulin and IGF/insulin in maintaining hESCs, our inhibitor experiments indicate that IGF/insulin signalling alone is sufficient.

Intriguingly, however, the authors suggest that hESCs have endogenous autocrine WNT signalling and that DKK1 (a WNT signalling inhibitor) further promoted hESC self-renewal[87]. Inhibiting WNT signalling also promotes naïve to primed hESC transition[88]. Given our differential gene expression analysis indicates WNT signalling differences in the EPI compared to AI-cultured hESCs, this suggests that modulating WNT may also

promote hESC self-renewal. The expression of JAK/STAT and NOTCH signalling components suggests that these pathways may also be operational in human embryos, and determining the role of these pathways may promote further refinement of human pluripotent stem cell culture media. It would also be interesting to determine if additional factors such as integrins contribute to maintaining pluripotency, both in the developing embryo and in hESCs. Although we find no major differences between AI-cultured hESCs on laminin compared to mTeSR1-cultured hESCs grown on either Matrigel or laminin, basement membrane signalling may be contributing to pluripotency maintenance in both contexts. Adding integrin blocking antibodies or activating

**Fig. 4 IGF1 treatment of human embryos promotes ICM proliferation. a** Representative western blots for selected proteins related to MAPK/PI3K/AKT/mTOR signalling in mouse and human embryos. $n = 6$ human blastocysts at 6 days post fertilisation (dpf) per experiment, and 5 or 10 mouse blastocysts. Given the limited human material, after WB transfer the higher and lower molecular weight regions of the blot were cut and probed separately. The cut junction can be seen in the upper edge of pS6. Full-length western blots are shown in Supplementary Fig. 11. Data represents $n = 5$ replicate experiments. **b** Schematic of human embryo treatment schedule. **c** Immunofluorescence analysis for NANOG (EPI, green) and SOX17 (PE, red) with DAPI (blue) nuclear stain at 6.5 dpf in control embryos or embryos treated with 17 nM IGF1 from 2 dpf. Scale bar: 100 µm. **d** Immunofluorescence analysis for NANOG (green), KLF17 (red) and DAPI (blue) at 6.5 dpf following 17 nM IGF1 treatment from 2 dpf. Scale bar: 100 µm. **e** Quantification of total cell number (DAPI stained nuclei) and percentages of NANOG- or SOX17-expressing cells, and ICM (combined NANOG and SOX17 totals) in control or 17 nM IGF1-treated embryos. Data points represent the percentage per individual embryo; horizontal line indicates the mean. $n = 5$ control, 4 treated embryos. One-tailed t-test. $*P < 0.05$, $**P < 0.01$. **f** Immunofluorescence analysis for NANOG (red) and GATA6 (PE, green), with DAPI (blue) at 6.5 dpf in control embryos or embryos treated with 1 µg/ml FGF2 and 1 µg/ml heparin, or 100 ng/ml FGF2 from 2 dpf. Scale bar: 100 µm. **g** Immunofluorescence analysis for NANOG (green) and SOX17 (red) with DAPI (blue) nuclear stain at 6.5 dpf in control embryos or embryos treated with 100 ng/ml FGF2 from 5 dpf. **h** Quantification of total cell number (DAPI) and percentages of NANOG- or SOX17-expressing cells, and ICM in control embryos or embryos treated with 100 ng/ml FGF2 from 5 dpf. Data points represent the percentage per individual embryo; horizontal line indicates the mean. $n = 10$ control, 9 treated embryos. One-tailed t-test. $*P < 0.05$, $**P < 0.01$. Source data are provided as a Source Data file.

integrins to each culture medium would provide insight into this question.

Altogether we demonstrate that taking an unbiased approach to interrogate lineage-specific gene expression patterns directly in the human blastocyst can be used to clarify the role of signalling in the embryo, and shed light on which pathways can be recapitulated in in vitro derived hESCs. Having defined a key signalling pathway required for the proliferation of early human embryos and hESCs, our next goal is to elucidate the pathways that regulate their pluripotency.

## Methods

**Ethics statement**. Human embryos were donated to the research project by informed consent under UK Human Fertilization and Embryo Authority (HFEA) License number R0162. Approval was also obtained from the Health Research Authority's Cambridge Central Research Ethics Committee, IRAS project ID 200284 (Cambridge Central reference number 16/EE/0067). The approval process entailed independent peer review along with approval from both the HFEA Executive Licensing Panel and the Executive Committees. Our research is compliant with the HFEA Code of Practice and has undergone independent HFEA inspections since the license was granted. Patient consent was obtained from Bourn Hall Clinic.

Informed consent was obtained from all couples that donated surplus embryos following IVF treatment. Before giving consent, donors were provided with all of the necessary information about the research project, an opportunity to receive counselling, and details of the conditions that apply within the license and the HFEA Code of Practice. Specifically, patients signed a consent form authorising the use of their embryos for research including stem cell derivation and for the results of these studies to be published in scientific journals. No financial inducements were offered for donation. Patient information sheets and the consent documents provided to patients are publicly available (https://www.crick.ac.uk/research/a-z-researchers/researchers-k-o/kathy-niakan/hfea-licence/). Embryos were donated cryopreserved and were transferred to the Francis Crick Institute where they were thawed and used in the research project.

**Human embryo culture**. Vitrified embryos frozen in straws were thawed by quickly transferring the contents of the straw from liquid nitrogen directly into thaw solution (Irvine Scientific Vitrification Thaw Kit) and thawed per manufacturer's instructions. Embryos frozen in cryopets were thawed for 3 seconds in a 37 °C water bath before transferring into thaw solution (Irvine Scientific Vitrification Thaw Kit). Embryos frozen in glass ampoules were thawed completely in a 37 °C water bath after the top of the vial was removed under liquid nitrogen. The contents were emptied onto a petri dish and the embryo transferred through a gradient of sucrose solutions (Quinn's Advantage Thaw Kit, Origio) per manufacturer's instructions.

Embryos were routinely cultured in Global Media supplemented with 5 mg/mL LifeGlobal Human Protein Supplement (both LifeGlobal) pre-equilibrated overnight in an incubator at 37 °C and 5% $CO_2$. For growth factor or inhibitor treatment, these conditions were supplemented with IGF1 (291-G1/CF, R&D) at a final concentration of 17 nM, or FGF2 (233-FB/CF, R&D) at a final concentration of 100 ng/ml, or at 1 µg/ml with 1 µg/ml heparin.

**Mouse embryo culture**. All mice were treated with appropriate care according to the ethics guidelines of The Francis Crick Institute. For mouse blastocysts, embryos at the two-cell stage were flushed out from the oviducts of C57BL6 × CBA F1 female mice that had been mated to C57BL6 × CBA F1 males. The embryos were cultured in vitro in KSOM (Millipore) until 3–4 days post fertilisation. All animal research was performed in compliance with the UK Home Office Licence Number 70/8560.

**hESC and iPSC culture**. For culture in knockout serum replacement (KSR) medium supplemented with FGF2 (KSR + FGF2), cells were maintained on mitotically inactivated MEF-coated pre-gelatinised tissue culture plates (Corning) in 20% KSR and 5 ng/ml of FGF2 (R&D). Cells were passaged by manual picking.

For culture in mTeSR1 medium (StemCell Technologies), cells were generally maintained on Matrigel-coated (BD Biosciences) tissue culture plates. Matrigel coating was performed for one hour at room temperature (RT) per the manufacturer's instructions. Cells were passaged as clumps using ReLeSR (Stemcell Technologies). ReLeSR was added to wells for 30 s at RT then aspirated, then plates were incubated for 5 min at 37 °C, quenched with mTeSR1 and lightly tapped to dislodge clumps of the desired size. Cells were also adapted to plates coated with 0.5 µg/cm² laminin-511 (Biolamina LN-511, Takara iMatrix-511 T303). Laminin coating was performed either overnight at 4 °C or for one hour at 37 °C per manufacturer's instructions. Cells on laminin-511 were also passaged with ReLeSR with a 7-min incubation at 37 °C, or dissociated to single-cell level with Accumax (Fisher Scientific), with a 10-min incubation at 37 °C.

For culture in TeSR-E8 medium (StemCell Technologies), cells were maintained on vitronectin-coated (StemCell Technologies) tissue culture plates. Vitronectin coating was performed for 2 h at RT per manufacturer's instructions. Cells were passaged as clumps using 0.5 mM EDTA. EDTA was added to wells for 5 min at RT then aspirated, then TeSR-E8 added to wells and cells disaggregated with a 5 ml stripette.

For culture in AI medium, cells were maintained on plates coated with 0.5 µg/cm² laminin-511. AI medium is made up as follows: first, make basal medium by combining 490 ml Advanced DMEM/F12 (Life Technologies 12634-010) or Essential 6 (ThermoFisher A1516401); 5 mL Penicillin Streptomycin (Gibco, 15140-122) and 5 mL GlutaMax (Life Technologies, 35050-038) and filter through a 0.22 µm bottle-top vacuum filter system. Basal media can be stored for up to one month in 4 °C. To make complete medium, add 50 ng/ml recombinant Human/Murine/Rat Activin A (Qkine, Qk001 or Peprotech, 120-14) and 17 nM IGF1 (291-G1/CF, R&D) to basal medium. Once growth factors are added, use media within 1 week. IGF1 was reconstituted to 200 µg/ml in sterile 1x $Ca^+/Mg^+$-free PBS. Activin A was reconstituted to 500 µg/ml in embryo culture grade water. Reconstituted growth factors were aliquoted and stored at −80 °C to minimise freeze-thaw cycles before use.

Early passage newly derived hESCs in AI medium were passaged by manual picking. Established cells in AI medium were also split using various dissociation reagents. For ReLeSR, cells were passaged similar to cells in mTeSR1 on laminin-511. For Accutase (StemCell Technologies) or Accumax, dissociation reagent was added to wells for 5–10 min, the well contents were collected and pelleted, and the dissociation reagent removed. Cells were resuspended in AI medium with or without 10 µm ROCK inhibitor (Y27632, Tocris) for plating. For Gentle Cell Dissociation Reagent (StemCell Technologies), reagent was added to wells for 6 min, then aspirated and AI medium added. Well contents were collected and disaggregated with a 5 ml stripette. For 0.5 mM EDTA (Sigma), reagent was added for 5 min then aspirated, AI medium added and cells disaggregated. For $Ca^+/Mg^+$-free PBS (Gibco, Life Technologies), reagent was added for 6 min, aspirated, then AI medium added and cells disaggregated.

For inhibitor treatment experiments, culture medium was supplemented with 100 nM PD173074 (FGFR tyrosine kinase inhibitor), 1 µm PD0325901 (MEK inhibitor), 10 or 1 µm OSI-906 (IGF1R/IR inhibitor), 100 nM GDC-9041 (PI3K inhibitor), 50 nM Everolimus (mTOR1 inhibitor), 1 mM CHIR99021 (GSK3β inhibitor) or 10 µm SB-431542 (Activin/Nodal receptor inhibitor). Cells were exposed to inhibitors for 24 h for western blot and 72 h for proliferation assays.

Naïve hESCs were cultured in t2iL + Gö[39] and maintained on mitotically inactivated MEF-coated pre-gelatinised tissue culture plates (Corning). Experiments were performed following one passage onto Geltrex (ThermoFisher, A1413301) coated tissue culture plates. Cells were passaged using Accutase. The dissociation reagent was added to wells for 5–10 min, the well contents were collected and pelleted, and the dissociation reagent removed. Cells were resuspended in t2iL + Gö medium with 10 μm ROCK inhibitor (Y27632, Tocris) for plating and ROCK inhibitor was subsequently removed for culture and during experiments.

**Immunofluorescence**. Embryos were fixed with 4% (w/v) paraformaldehyde (PFA) (ThermoFisher) for 1 h at 4 °C on a rotating shaker, then transferred through several washes of 0.1% (v/v) Tween-20 (Sigma) in Ca+/Mg+-free Dulbecco's Phosphate-Buffer saline (PBS) without calcium and magnesium (ThermoFisher) to remove residual PFA. Embryos were permeabilised in PBS 0.5% Tween-20 for 20 min, then transferred to blocking solution (10% (v/v) donkey serum (Sigma) in PBS 0.1% Tween-20) for 1 h at RT. Embryos were placed in primary antibodies (Supplementary Table 1) at a 1:500 dilution in blocking solution overnight at 4 °C on a rotating shaker. Embryos were then transferred through four washes of PBS 0.1% Tween-20 followed by a final 30 min wash. Embryos were placed in secondary antibodies (Supplementary Table 2) at a 1:300 dilution in blocking solution for 1 hour at RT on a rotating shaker, then transferred through four washes of PBS 0.1% Tween-20 and a final 30 min wash. Embryos were placed in a 50 μL 1:30 dilution of DAPI-containing Vectashield (Vector Labs) in PBS 0.1% Tween-20 on a coverslip bottom dish (MatTek) for confocal imaging.

hESCs, iPSCs and differentiated cells were subjected to a similar protocol. Briefly, cells were washed twice with Ca+/Mg+-free PBS and fixed with 4% (PFA) for 1 h. Cells were then permeabilised with PBS 0.5% (v/v) Triton X-100 (Sigma) or PBS 0.5% Tween-20, then transferred to blocking solution (2–10% donkey serum in PBS 0.05% or 0.01% Tween-20), before overnight incubation at 4 °C with primary antibodies. Cells were washed three times with PBS 0.05% or 0.01% Tween-20, then secondary antibodies diluted at 1:300 in blocking solution were added for 1 hour at RT. Cells were washed several times in PBS 0.1% Tween-20 and placed in a final wash for 30 min. A drop of DAPI-containing Vectashield was added to each well prior to imaging.

Primary and secondary antibodies are listed in Supplementary Tables 1 and 2, respectively.

**Embryo confocal imaging and immunofluorescence quantification**. Embryos were imaged on a Leica SP5 inverted confocal microscope (Leica Microsystems GmbH) at a z-section thickness of 3 μm. Confocal stacks in TIFF format were loaded into the MINS 1.3 software automated pipeline to detect and segment nuclei and determine the number of cells in each embryo (http://katlab-tools.org/)[72]. The MINS segmentation output was manually checked for appropriate segmentation and mitotic nuclei were removed from the analysis.

**Imaging of cell lines**. Images were acquired using a Leica TCS SP8 confocal microscope or an Olympus 1 × 73 microscope with Cell^F software (Olympus Corporation). Phase-contrast images were taken on a Leica DM IL LED microscope with a Leica MC170 H9 camera (Leica Microsystems GmbH).

**hESC derivation from preimplantation embryos**. Human blastocysts at 5 or 6 days post fertilisation (dpf) were initially cultured in Global Media (LifeGlobal) supplemented with 5 mg/mL LifeGlobal Protein Supplement pre-equilibrated in an incubator at 37 °C and 5% CO₂ prior to stem cell derivation. Embryos at 6 dpf were microdissected to isolate the ICM using an Olympus IX73 microscope and a Saturn 5 laser (Research Instruments), as described previously[89]. Embryos were placed in drops of Global® Total® w/HEPES (LGTH, LifeGlobal) on a Petri dish overlaid with mineral oil for micromanipulation. The ICM and polar trophectoderm were plated onto mitotically inactivated MEF-coated dishes in AI medium. ICM outgrowths with hESC-like morphology were manually picked onto laminin-511 coated dishes for further propagation. Three independent hESC lines were established: CH1, CH2 and CH3.

**Human iPSC derivation**. Human BJ fibroblast cells (ATCC ® CRL-2522) were plated so as to be 30–60% confluent for transduction two days later (1 × 10⁵ cells per well of a six-well plate). Cells were transduced using the Cytotune™ 2.0 Sendai reprogramming kit (Invitrogen) per manufacturer's instructions, and transferred 6 days after transduction to dishes coated with either MEFs (KSR + FGF2 medium), vitronectin (TeSR™-E8 medium) or laminin-511 (AI medium). At this point cells were transferred to hypoxia conditions (5% O₂, 5% CO₂, 37 °C) for the remainder of the derivation. TRA-1-60 expression was analysed 18 days after transduction using the Stemgent® StainAlive™ TRA-1-60 Antibody (DyLight™ 488) kit, per manufacturer's instructions (1:100 dilution). Colonies with pluripotent ES cell morphology were picked 22 days after transduction for expansion to establish stable iPSC lines. Alkaline phosphatase staining was carried out using an Alkaline Phosphatase Detection Kit (SCR004, Millipore) as per manufacturer's instructions.

Human MRC5 cells (ECACC 05072101) were routinely cultured in DMEM (Sigma D6546) supplemented with 10% FBS (Sigma, F7524) and 2 mM Glutamax

(Gibco®). Cells were plated on laminin-511 at 1 × 10⁵ cells per well of a six-well plate one day prior to reprogramming in MRC5 media. On the day of reprogramming media was changed to AI medium or Nutristem hPSC XF culture medium (Stemgent 01-0005). Cells were reprogrammed over 4 days using the Stemgent® StemRNA™ 3rd Gen Reprogramming Kit for Reprogramming Adult and Neonatal Human Fibroblasts (Reprocell, 00-0076) as per manufacturer's instructions. Cells were maintained under the following incubator conditions: 5% O2, 5% CO2, 37 °C. From day 5 onwards, daily complete media changes were performed with either AI or Nutristem media. Live-cell staining with alkaline phosphatase (ThermoFisher, A14353) was performed as per manufacturer's instructions on day 34following the first of four transfections.

**Cell proliferation assay**. Cells cultured in AI, mTeSR1, KSR + FGF or t2iL + Gö medium were plated on day 0 at $2 × 10^4$ cells/cm². Cell counts were measured every day thereafter for 3 or 4 days using a Nucleocounter NC-200 (Chemometech) or a Scepter Automated Cell Counter (Millipore). Population doublings were calculated according to Equation 1 or cell density was plotted.

$$PDL = \frac{\log_{10}\frac{\text{Viable cell number at time } t}{\text{Viable cell number at time } t-1}}{0.30103}$$

**Flow cytometry**. Expression of surface and intracellular pluripotency-related proteins was analysed using a MACSQuant® Analyzer10 flow cytometer (Miltenyi Biotech) or BD LSRFortessa™ flow cytometer (BD Biosciences). Single-cell suspensions were washed and stained in BD Pharmigen™ Stain Buffer (BD Biosciences) with conjugated primary antibodies (Supplementary Table 1). For intracellular proteins (NANOG, OCT4, SOX2), cells were fixed using BD Cytofix™ fixation buffer (BD Biosciences) and incubated for 15 min at RT. Cells were then permeabilised with 0.1% Triton X-100 (Sigma) in BD Pharmigen™ Stain Buffer for 15 min at RT prior to staining. Isotype controls were performed for each antibody (listed in Supplementary Table 3). Cells were stained with Live/Dead® discrimination dye (L23105, ThermoFisher) and phenotype analysis of the live single-cell population fraction performed by flow cytometry. Isotype staining was considered as a negative control for each analysis and condition.

**hESC targeting in AI media**. hESCs were cultured on laminin-511 in AI media and pre-treated for 16 h with 10 μM Y-27632 (Tocris). hESCs were dissociated to single cells with Accumax (Thermo Fisher Scientific), with a 10 min incubation at 37 °C. $2 × 10^6$ cells were nucleofected in 100 μl with a total of 4 μg of either the targeting vector (gRNA targeting *ARGFX* cloned into pSpCas9(BB)-2A-Puro [pX459] v2.0) or pMaxGFP (supplied with the kit) using the Lonza P3 Primary Cell 4D-Nucleofector X Kit and the cycle CB 150 on a Lonza 4D-Nucleofector System, according to the manufacturer's instructions. Nucleofected hESCs were plated onto three wells of a six-well plate on a feeder layer of DR4 MEFs (puromycin and neomycin resistant) and cultured in AI media (without antibiotics) containing 10 μM Y-27632. After 30 h of culture the cells were selected in AI media containing 0.5 μg/ml of puromycin for 48 h. Following selection, the colonies were allowed to grow for 10 days, after which DNA was isolated from one well of the 6-well plate and subjected to the T7 endonuclease I assay to assess targeting efficiency as described in Renouf et al.[90]. The remaining 2 wells were subjected to alkaline phosphatase stain as described above.

pSpCas9(BB)-2A-Puro [pX459] v2.0 was a gift from Feng Zhang (Addgene plasmid#62988)

The gRNA sequence used for ARGFX: CCTACCGGAGTCAACAGTAA.

The primers used for the amplicon PCR are:

Forward primer: GAAGCAATACGGAGAAGGCA

Reverse primer: GGGTAGAGGGTGGGGAATTT

**Karyotype by G-banding and whole-genome sequencing**. For G-banding karyotype analysis, hESCs and iPSCs were fixed in suspension. Multiple metaphase spreads were analysed per sample and the number of chromosomes and G-banding pattern were determined. For whole-genome sequencing a DNeasy Blood & Tissue kit (Qiagen) was used to isolate genome DNA from hESCs cultured in AI medium. Isolated DNA was quantified on a Nanodrop prior to library preparation using a Nextera XT kit (Illumina). Libraries were sequenced paired-end with at least 100 bp reads on a HiSeq 4000 machine (Illumina).

**RNA FISH**. RNA FISH was carried out as previously described[91] with minor modifications. Two drops of cell suspension in phosphate-buffered saline were added on to a slide placed on cold platform and cells were allowed to settle for 3–4 min. Cells were permeabilised with ice-cold 0.5%Triton X-100, 2 mM Vanadyl Ribonucleoside (Biolabs) in PBS for 10 min and fixed with ice cold 4% paraformaldehyde (pH 7–7.4) for 10 min. Slides were rinsed in PBS, dehydrated through an ethanol series (2 × 70%, 80%, 96%, 100%) and air-dried. BAC DNA was labelled using Nick Translation Direct Labelling Kit (30-608364/R1; Roche) according to manufacturer's instructions, and using fluorescent nucleotides spectrum orange (02N33-050, Vysis)—dUTP for *XIST* (BAC-RP11-13M9) and spectrum green (02N32-050, Abbott)—dUTP for *ATRX* (BAC-RP11-1145J4). Cells were hybridised with a denatured mix of probes (*XIST* 0.1 μg probe and *ATRX* 0.1 μg probe), 3 μg

human COT1 DNA and 10 μg salmon sperm DNA in hybridisation buffer (50% formamide, 25% dextran sulphate, 5 mg/ml BSA, 1 mM Vanadyl Ribonucleoside complex in 2X SSC) at 37 °C overnight in a humid chamber. Slides were subjected to stringency washes at 42 °C, three times for 5 min in 50% formamide, 1X SSC (pH 7.2–7.4), and three times for 5 min in 2X SSC (pH 7–7.2). Slides were mounted in antifade with DAPI.

**Differentiation assays**. For spontaneous differentiation, hESCs cultured in AI medium were switched into MEF culture medium (10% FBS) for 6 or 12 days, and cells fixed for immunofluorescence analysis for germ layer lineage markers.

For directed differentiation to hepatocytes, hESCs adapted to AI medium were taken through a previously described protocol[54]. hESCs were dissociated to single cells using Accutase then resuspended in AI medium containing 10 μm ROCK inhibitor for seeding at $1 \times 10^5/cm^2$ onto gelatinised tissue culture plates pre-coated with MEF medium. AI medium was replenished 24 h later. The hepatocyte differentiation protocol was started 48 h after seeding, and cells were maintained under hypoxia conditions from then on. Cells were fixed for immunofluorescence analysis 3 days (endoderm), 8 days (foregut endoderm) and 25 days (mature hepatocytes) following differentiation induction.

For directed differentiation to cardiomyocytes, hESCs were cultured using the STEMdiff Cardiomyocyte Differentiation Kit (StemCell Technologies, 05010). Cells adapted to AI medium were seeded at a density of $1.625 \times 10^6/cm^2$ onto Matrigel-coated plates, with 10 μm ROCK inhibitor. AI medium was replenished 24 h later. From day 2, cells were treated according to the manufacturer's protocol: two days with medium A, two days with medium B, and four days with medium C. From 8 to 15 days after differentiation, cells were fed with the provided cardiomyocyte maintenance medium (replenished every 2 days). RNA samples were collected 8 and 15 days following differentiation induction, and cells fixed for immunofluorescence at day 15.

For neuronal progenitor cells, hESCs adapted to AI medium were differentiated using a modification of previously described protocols[55,92]. Cells adapted to AI medium were seeded at a density of $1.5 \times 10^6/cm^2$ onto laminin-coated plates, with 10 μm ROCK inhibitor. AI medium without ROCK inhibitor was replenished 24 h later. From day 2, confluent cells were treated with N2B27 medium supplemented with 10 μm SB431542 and 1 μm LDN193189 (Stemgent). On days 7, 12 and 16 following induction, cells were passaged with Accutase for 4 min, cell collected in N2B27 media, spun down and then plated in either N2B27 medium supplemented with 10 μm SB431542 and 1 μm LDN193189 (until day 12) or N2B27 media (from day 12 to 20) for subsequent culture. Cells were fixed for immunofluorescence analysis 7, 12, 16 and 20 days following differentiation induction.

**Quantitative RT-PCR**. RNA was isolated using TRI Reagent (Sigma) and DNaseI treated (Ambion). cDNA was synthesised using a Maxima First Strand cDNA Synthesis Kit (Fermentas). Quantitative RT-PCR (qRT-PCR) was performed using Sensimix SYBR Low-Rox kit (Bioline, QT625) on a QuantStudio 5 machine (ThermoFisher). Primer pairs (listed in Supplementary Table 4) were previously published[93] or designed using Primer3 software.

**Western blot**. For hESCs and iPSCs, protein was extracted with CelLytic M reagent (Sigma) supplemented with protease and phosphatase inhibitors (Roche) and 15–30 μg of hESC or iPSC protein was resolved on 10% SDS-PAGE gels. For mouse and human embryos, samples were washed three times with $Ca^+/Mg^+$-free PBS and were directly lysed in Laemmli sample buffer at 95 °C for 5 min. Between 2 and 10 embryos were included per sample as indicated, and resolved on 10% Mini-Protean TGX precast protein gels (BioRad, 4561036).

Resolved protein gels were transferred to a PVDF membrane using a BioRad Trans-Blot transfer system (BioRad). Membranes were blocked in TBS and 0.1% Tween (Sigma) with 5–10% skimmed milk, and incubated with primary antibody at 4 °C overnight. Following washes in TBS 0.1% Tween, membranes were incubated with secondary antibody for one to 3 h at RT. Proteins were visualised using the Pierce ECL Western Blotting Substrate (ThermoFisher) using an Amersham Imager 600. Antibodies are listed in Supplementary Tables 1 and 2. Full-length western blots are shown in Supplementary Fig. 11.

**Single-cell RNA-seq**. cDNA was generated from single cells using the SMART-Seq V4 Ultra Low Input Kit (Clontech 634898) as previously described[10]. cDNA was amplified by adding 25 μl 2X SeqAmp PCR buffer, 1 μl PCR Primer II A (12 μM), 1 μl SeqAmp DNA polymerase, 3 μl nuclease-free water, and incubated on a PCR machine at 95 °C for 1 min, 23 cycles of 98 °C for 10 s, 65 °C for 30 s and 68 °C for 3 min, before a final extension for 10 min at 72 °C. Amplified cDNA was purified using 50 μl AMPure XP beads (Beckman Coulter) per manufacturer's instructions. Sample tubes were mixed well and incubated at RT for 8 min to bind the cDNA to the beads. Tubes were put on a magnet until the supernatant ran clear so that it could be removed and discarded. The beads were washed twice with 200 μl 80% ethanol. Any remaining ethanol was removed and beads allowed to dry, then resuspended in 12 μl of elution buffer. Tubes were returned to the magnet and the clear supernatant containing the cDNA was collected in a new tube and stored at −80 °C until library preparation.

cDNA quality was assessed by High Sensitivity DNA assay on an Agilent 2100 Bioanalyzer, with good quality cDNA showing a broad peak from 300 to 9000 bp. cDNA concentration was measured using Qubit dsDNA HS kit (Life Technologies).

In preparation for library generation, 10 μl of cDNA sample and 32 μl purification buffer were added to a Covaris microTUBE AFA Fiber Pre-Slit Snap Cap. cDNA was sheared to a 200–500 bp range using an E220 focused-ultrasonicator (Covaris) for 2 min at Peak Incident power 175 W, Duty Factor 10%, 200 cycles per burst, water level 5. Libraries were prepared using a Low Input Library Prep Kit HT (Clontech, 634900) per manufacturer's instructions. Library quality and cDNA concentration was assessed by Bioanalyzer and Qubit. For library purification, 75 μl of AMPure XP beads was added to each collection tube containing the cDNA and treated as described above.

**Analysis of transcriptome datasets**. A detailed analysis pipeline can be found at the following site: https://github.com/galanisl/AI_hESCs.

Single-cell RNA-seq samples generated in this study were compared to published RNA-seq datasets for hESCs and human embryo EPI, PE and TE cells[10,28,31,33,37,56,58]. Read quality control was performed using FastQC v0.11.6 and results integrated with multiqc v0.9. Quality and adapter trimming with Trim Galore! v0.5.0 were applied to samples that did not pass the quality controls. Mapping was performed against the human reference transcriptome GRCh38 using the salmon v0.11.3 program[94]. Transcript IDs were mapped to gene IDs and official gene symbols using biomaRt v2.36.1. Gene-level summarisation of counts and combination of all sample data was carried out with tximport v1.8.0.

The tximport list was transformed into a DESeqDataSet with DESeq2 v1.20.0[95]. Mitochondrial-, ribosomal- and pseudo-genes were removed from the count matrix, as well as no-show and invariant genes. We normalised the count data by size factor using DESeq2 to remove the correlation between average counts of a gene across samples and variance. We then log-transformed the normalised expression matrix (Supplementary Data 6). In addition, we removed batch effects due to sequencing type (bulk vs. single-cell) using the limma v3.36.5 program[96]. Briefly, the function removeBatchEffect in limma fits a linear model to the data, including batches and the condition of interest, and then removes the component due to the batch effects. We imputed drop-out events in the samples sequenced with single-cell RNA-seq via DrImpute v1.1[97]. To identify highly variable genes (HVGs) for dimensionality reduction of the size-factor-normalised data, it was assumed that the trend fitted to the mean-variance curve represented the technical component of the variance, with expression values that significantly deviate from this reference corresponding to HVGs[98]. The functions trendVar() and decomposeVar() from scran v1.8.4[98] were applied, which identified 3087 highly variable genes (HVGs) at the significance level $\alpha = 0.01$ after Benjamini-Hochberg correction.

Gene expression data was also embedded to three-dimensional space using UMAP[57], a non-linear dimensionality reduction approach. We used $k = 30$ nearest neighbours to construct the initial proximity graph for the human embryonic cells and ESCs and $k = 15$ when the cynomolgus monkey data was integrated.

The griph tool (https://github.com/ppapasaikas/griph) was used to identify clusters of transcriptionally similar cell types, accounting for confounding factors such as the cell cycle stage or batch effects to identify clusters of transcriptionally similar cell types.

Genes identified as differentially expressed from a DESeq2 analysis were used to perform a Gene Ontology[60] and REACTOME[61] functional enrichment analysis. The FunEnrich package (https://github.com/galanisl/FunEnrich) was used with focus on the five most enriched GO terms and REACTOME pathways based on Benjamini-Hochberg corrected p-values.

To integrate the RPM-normalised gene expression matrix from the cynomolgus monkey samples[99] with human ESC and embryo data, we normalised the latter by RPM and filtered all datasets in order to keep only orthologs between human and monkey[59]. Orthology mapping was based on published gene lists[99]. Furthermore, we removed no-show genes for dimensionality reduction.

**Gene expression analysis**. We identified pathways enriched in the Epi versus the PE and TE, as well as the TE versus the PE using gene set expression analysis[38]. For this, we used a compendium of multiple pathway databases compiled by the Bader Lab at http://download.baderlab.org/EM_Genesets/current_release/Human/symbol/Pathways/ (01/08/2019 snapshot). Genes were ranked according to log2-fold changes between each pairwise comparison to generate rank files that were then used as input for the GSEAPreranked module of the GSEA v3.0 tool. We scanned the output files with v3.2.0 of the EnrichmentMap app[39] in Cytoscape[40] v3.7.1 to represent the relationship between enriched signalling pathways using a significance level of 0.05 for the nominal p-values.

The list of ligands that activate the MAPK signalling pathway was extracted from the corresponding entry (hsa:04010) on the KEGG pathway database[41].

To identify ligand-receptor partners, we manually extracted a list of all ligands and receptors from all signal transduction pathways of the KEGG database[41]. We paired ligands with their corresponding receptors based on protein interaction data from version 2.2 of the HIPPIE database[42]. Finally, we calculated the median expression of ligands and receptors in the epiblast, primitive endoderm and trophectoderm using RPKM expression values.

We used version 1.22.3 of the R package pathview[100] to overlay log2-transformed median RPKM expression values on all signal transduction pathways

of the KEGG database[41]. For pathway members that represent gene categories instead of a single gene, we used the maximum expression of all the genes falling in the category. A range 0 to 15 in the log2-transformed RPKM scale was used for the three embryonic cell types and the window was divided in 20 bins to define colour intensities for the pathway members.

**Shiny app**. We used version 1.3.2 of the R package shiny to develop a Shiny App to facilitate access to the above KEGG pathway diagrams via an interactive web page. This Shiny App colours pathway members according to their expression in the three different cell types of the human blastocyst (epiblast, primitive endoderm and trophectoderm). All the KEGG signal transduction pathways under the category Environmental Information Processing (see https://www.genome.jp/kegg/pathway.html#environmental) can be chosen from a dropdown menu, as well as selecting the cell type based on which the pathway members are coloured. Gene expression is shown as a colour range between 0 and 15 in log2 (RPKM + 1) units divided in 20 bins. Since gene expression comes from a single-cell RNA-seq dataset[10,37], pathway members on the KEGG diagram are coloured with the median expression of the gene across single cells of the same type. When pathway members represent a gene category instead of a single gene, the maximum median is represented. Clicking on pathway members generates boxplots with the expression distribution of the gene or genes in the three blastocyst cell types.

**Analysis of low-pass whole-genome sequencing**. We performed quality control with FastQC v0.11.8 and adapter and quality trimming with Trim Galore! v0.6.0 if needed. We aligned samples to the UCSC hg19 reference genome using the two-pass alignment pipeline of STAR v2.7.0d. Copy number analysis of the samples was performed using the R package QDNAseq v1.18.0. With this package, we divided the hg19 reference genome into non-overlapping bins of size 1000 kb. Then, we counted the number of sequence reads per bin and used these counts to estimate copy numbers. These estimates are normalised and outliers smoothed. Finally, we performed segmentation and plotted the copy number profile.

**Reporting summary**. Further information on research design is available in the Nature Research Reporting Summary linked to this article.

## Data availability
The authors declare that all data supporting the findings of this study are available within the article and its supplementary information files or from the corresponding author upon reasonable request. RNA-seq FastQ files have been deposited in the Gene Expression Omnibus repository under accession code GSE126488 and using the IDs shown at https://github.com/galanisl/AI_hESCs. The source data underlying Figs. 2d, 3c, 4e, h, Supplementary Figs. 5h, d and 7a are provided as a Source Data file.

## Code availability
A detailed analysis pipeline can be found at the following site: https://github.com/galanisl/AI_hESCs

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

## Acknowledgements

We thank the generous donors whose contributions have enabled this research. We are grateful to the members of the K.K.N., J.M.A.T. and Robin Lovell-Badge labs for their helpful advice and comments on the paper. We thank Ricardo Baptista, Iris Valero and Terri Gaskell of the Cell and Gene Therapy Catapult for independent validation of AI medium using hESC and iPSC lines; Donald Bell for expert imaging advice and for imaging the reprogrammed iPSCs; Sukhveer Purewal for assistance with flow cytometry analysis; Jim Smith, Andrea Bernardo and Fay Cooper for their advice on cardiomyocyte differentiation and antibodies; Ludovic Vallier and Carola Morell for their advice on hepatocyte differentiation; Francois Guillemot and Oana Paun for their advice on neural progenitor cell differentiation and antibodies; the Francis Crick Institute's Science Technology Platforms: Advanced Light Microscopy, High Throughput Sequencing, and Bioinformatics; Amy Strange and Marc Pollitt from the Francis Crick Institute's Digital Team for their help with the Shiny App deployment; and Veronique Birault, Ranmali Nawaratne and Isabel Ramos in the Francis Crick Institute's Translation Group. J.D. is supported by a Wellcome Trust Senior Investigator Award 103799/Z/14/Z. Work in the K.K.N., J.M.A.T. and J.D. labs was supported by the Francis Crick Institute, which receives its core funding from Cancer Research UK (FC001120, FC001193 and FC001070), the UK Medical Research Council (FC001120, FC001193 and FC001070), and the Wellcome Trust (FC001120, FC001193 and FC001070). Work in the K.K.N. laboratory was also supported by the Rosa Beddington Fund and Medical Research Council's Confidence in Concept Crick i2i funding (MC_PC_16062 10609).

## Author contributions

Most of the experiments were performed by S.E.W. and K.J.G., with assistance from A.M., C.G., S.O., R.L., S.K.M., L.H. and L.D. G.A.-L. performed all the computational analysis. M.M.-A. and J.D. provided key reagents and advice on inhibitor experiments. P.S., L.C. and K.E. assisted with the consent and donation of human embryos for research. J.M.A.T. provided key reagents and advice. K.K.N. and S.E.W. conceived the study, analysed data, and wrote the paper with input from all of the authors.

## Competing interests

The Francis Crick Institute has filed a patent application (WO2018130831) relating to a Composition for Culture of Pluripotent Stem Cells (K.K.N. and S.E.W.). All other authors declare no competing interests.
