## [Peer Review File · Nature Communications]

Reviewers' Comments:

Reviewer #1:

Remarks to the Author:

In this manuscript, the authors analyse datasets from single cell transcriptomic of human blastocysts to unbiasedly mine the signalling pathways that are active in the Epi compartment. Beyond the classical Nodal pathway (that they previously assessed), the authors point at the IGF1/IGFR1/PI3K/AKT pathway. Most intriguingly, this dataset does not pinpoint a role for the FGFs, which is largely exploited by current hESC culture protocols. Based on this finding, they define new culture conditions (IGF1/Activin/L511) for hESCs that more closely resemble the in vivo situation. This novel culture condition allows for the culture of established lines and for the establishment of de novo lines from the blastocyst and via reprogramming. These lines are competent for differentiation into hepatocyte-like and cardiomyocyte-like cells. This novel state is similar to the primed state of pluripotency both at the transcriptome level and regarding the X chromosome inactivation status. The authors then show convincing data showing that the PI3K/AKT pathway regulates the pluripotent state both in hESCs and in the Epi of the blastocyst. Overall, this study establishes a novel systematic and rational way to investigate the regulation of human pluripotency and defines a novel culture condition that recapitulates that state in vitro. It also reveals that the cross-talk between the AKT and ERK pathway explains the previous misunderstanding and seemingly inappropriate use of FGFs to maintain the human pluripotent state in vitro.

Line 131: Please add a reference to the statement 'the interleukin 6 receptor subunit IL6R, which can bind insulin-related ligands.'

Line 135 and Supplementary figure 1b:

- Provide the full list of all genes, Log2foldchange, value, padj, basemean, lfcSE, and stat for Epi enriched, TE enriched, and PE enriched in a XCel tab, to complement SF1a.
- Provide the plots for the expression levels of all ligands known to activate the MAPK pathway including but not restricted to PDGF, EGF, HB-EGF, FGF ligands in Epi, TE, PE, even if they are lowly- or non-expressed.
- Provide insights into the regulation of downstream proteins relative to GPCR pathways that are known to regulate the MAPK pathway (e.g. but not restricted to PKA, PKC).

Line 136: Beyond the factors secreted by the EPI, the pluripotency might be maintained by niche factors produced by the TE and PE. To challenge your finding that IGF1 is an important regulator of human pluripotency, the authors should provide the lists and plots of all secreted ligands (a.k.a. secretome) and of the corresponding receptors produced by the Epi, TE, and PE. This data would be especially relevant to assess the putative role of TGF β , WNT, and STAT pathways.

Line 136: Because FGFR4 is expressed in the Epi, the authors should use REACTOME or KEGG analysis to assess the regulation of classical MAPK/ERK target genes in the Epi as compared to the TE and PE. This analysis should be extended to the WNT, TGF β , and STAT pathways that were proposed to also regulate human pluripotency.

Line 156: Integrins and FAK have been associated with MAPK activity (e.g., DOI: 10.1038/372786a0). The authors should test whether the L511 regulates MAPK activity, for example as compared to Matrigel and other laminins previously used for hESC culture. This might in part explain the differences in ligands and signalling activity that they observed. Other controls including the addition into the culture medium of antibodies blocking or activating integrins might be insightful as well.

Line 180: The authors should add controls to their FACS experiment including hESCs cultured in F2/KSR and t2iLGö.

Line 184: The putative superiority of AI over other culture conditions for clonal derivation should be quantified via a CFU assay.

Line 190: The author should clearly mention in the conclusions that they used multiple lines and name them (H1, H9, CH2, Shef6).

Line 200: The authors should mention the number of cell lines that they derived do novo. Was the derivation efficiency similar to FGF-driven derivation? Did the authors see reproducible differences

in the transcriptome of de novo lines as compared to converted lines? This might point at irreversible changes due to a derivation in FGF as compared to IGF1.

Line 258: Due to the sensitivity of the adjustment of batch effects, it would be important to briefly mention the method in the text.

Line 287: The pair-wise analysis should be also done using Panther and KEGG in order to depict the differences in signalling activity.

Line 293-298: The lower WNT activity in Epi as compared to AI- and TeSR-cultured hESCs is very interesting as it contradicts the current paradigm that WNT should be activated in naive hESCs. The enrichment for genes associated with IL6 activity in both Epi and AI-hESCs is very interesting as well. It is thus tempting to ask the authors to perform longer-term culture of hESCs in AI supplemented with WNT inhibitors (e.g., IWP2, XAV939), PD17/PD03 (in relation to line 371 and 382, and possibly at concentration 0.5 μ M for PD03, in light of doi:10.1038/s41592-018-0104-1) and in the presence of IL6.

Line 351: The authors should check whether the elements of the mTOR pathway are differentially regulated in the Epi as compared to the TE and PE.

Line 428: The authors should use their refined WB technique to check the levels of pERK in human blastocysts as compared to mouse blastocysts (in which the pathway activity is well established) and to hESCs in AI-, KSR-, and t2iLGö-culture conditions. The hESCs samples can be dosed from 1 to 100 cells for example, in order to give an idea of the respective levels.

As a summary, in order to make this paper a landmark in hESC culture by unbiasedly assessing signalling pathways regulating human pluripotency, I would suggest that the authors:

- perform additional computational analysis of the single cell sequencing data to:
 - (i) provide the full list of all genes, Log2foldchange, value, padj, basemean, lfcSE, and stat for Epi enriched, TE enriched, and PE enriched,
 - (ii) provide the plots for the expression levels of all ligands activating the MAPK pathway including but not restricted to PDGF, EGF, HB-EGF, FGF ligands in Epi, TE, PE,
 - (iii) provide insights into the downstream proteins of GPCR pathways that are also known to regulate the MAPK pathway (e.g., PKA, PKC)
 - (iv) provide the list and plots of all secreted ligands and of the corresponding receptors produced by the Epi, TE, and PE, especially for the TGFB, WNT, and STAT pathways
 - (v) provide panther and KEGG analysis of the AI- as compared to F2/KSR- and t2iLGö-cultured hESCs
- perform an additional experiment to weight the induction of MAPK activity via integrins
- add controls to the FACS experiment (Fig 1c) to quantify and compare the levels of pluripotency-related genes
- test the clonal efficiency of AI- as compared to F2/KSR- and t2iLGö-cultured hESCs
- attempt the long-term culture and characterise AI-hESCs in the presence of WNT inhibitors (IWP2, XAV939), ERK inhibitors (PD17, PD03) and IL6 [modified AI culture]
- verify the levels of pERK in human blastocysts as compared to mouse blastocysts and to multiple doses of hESCs in AI-, modified AI-, F2/KSR- and t2iLGö-cultured hESCs.

Reviewer #2:

Remarks to the Author:

This manuscript shows that FGF2 is dispensable in the culture of human embryonic stem cells (hESCs) when replaced with IGF1 and insulin. This conclusion is based on a screen from prior single cell RNA seq data of a day 6 embryo and based on their work culturing hESCs in IGF1-containing media. I think their results are interesting and will be valuable to the field. My main concern, however, is that this result is not very novel. They show that cells in AI medium have a highly similar transcriptomics profile to hESCs conventionally grown in mTsr (with the exception of morphology). At the authors' own admission, "IGF1 and FGF2 may activate overlapping downstream mechanisms for the maintenance of hESCs". Cells cultured in these two media also

respond similarly to differentiation cues. Based on that, it is unlikely that the altered culture conditions will change our current hESC media practices and the value of the paper is in linking the IGF pathway to pluripotency.

However, this point is not novel either. The authors missed previous references that have already shown that PI3K/AKT signaling is active in hESCs, that it is necessary for pluripotency maintenance and that this signaling is activated by insulin/IGF (McLean et al. 2007; Singh et al. 2012). Furthermore, FGF-free media were also developed and used to maintain pluripotency. It included heregulin in addition to IGF and Activin (Singh et al. 2012). Cross-talk between PI3K and MAPK pathways in hESCs has also been previously demonstrated at the level of pAKT and pERK (Singh et al. 2012). The authors should discuss and cite this prior work. Also given the findings of these papers, I think all instances of the word "first" should be removed from the current paper.

Using the human embryo to optimize the culture conditions for hESCs is a novel approach with a strong rationale. However, the fact that AI culture conditions maintain pluripotency is not surprising given the above-mentioned work. The main new insight that the authors add is to demonstrate that the FGF receptor can be inhibited without consequences to pluripotency, thus providing clear evidence that FGF signaling is dispensable for hESC maintenance and that the main function of FGF in conventional media is to activate PI3K signaling.

Additionally, the authors should consider comparing transcriptome data for AI conditions to other media conditions in which cells are also grown on laminin-511, so as to highlight any differences between using FGF and IGF rather than differences resulting from the extra-cellular matrix. This would strengthen the paper and the impact of its findings. If this comparison was already made then the authors should discuss it in more detail (e.g. AI vs mTeSR1 on Laminin-511).

It is crucial to discuss these points and revise the paper accordingly before further consideration. In all, while I think the work is solid, my concern is that not very much about the work (culture conditions or the mechanism itself) is very novel and it is up to the editor to decide whether it is suitable for Nature Communications.

Missed references:

McLean, Amanda B, Kevin A D'Amour, Karen L Jones, Malini Krishnamoorthy, Michael J Kulik, David M Reynolds, Alan M Sheppard, et al. 2007. "Activin Efficiently Specifies Definitive Endoderm From Human Embryonic Stem Cells Only When Phosphatidylinositol 3-Kinase Signaling Is Suppressed." 25 (1). John Wiley & Sons, Ltd.: 29–38. doi:10.1634/stemcells.2006-0219.

Singh, Amar M, David Reynolds, Timothy Cliff, Satoshi Ohtsuka, Alexa L Mattheyses, Yuhua Sun, Laura Menendez, Michael Kulik, and Stephen Dalton. 2012. "Signaling Network Crosstalk in Human Pluripotent Cells: a Smad2/3-Regulated Switch That Controls the Balance Between Self-Renewal and Differentiation.." Cell Stem Cell 10 (3): 312–26. doi:10.1016/j.stem.2012.01.014.

Response to reviewers NCOMMS-19-16013-T

Reviewers' comments:

Reviewer #1 (Remarks to the Author):

In this manuscript, the authors analyse datasets from single cell transcriptomic of human blastocysts to unbiasedly mine the signalling pathways that are active in the Epi compartment. Beyond the classical Nodal pathway (that they previously assessed), the authors point at the IGF1/IGFR1/Pi3K/AKT pathway. Most intriguingly, this dataset does not pinpoint a role for the FGFs, which is largely exploited by current hESC culture protocols. Based on this finding, they define new culture conditions (IGF1/Activin/L511) for hESCs that more closely resemble the in vivo situation. This novel culture condition allows for the culture of established lines and for the establishment of de novo lines from the blastocyst and via reprogramming. These lines are competent for differentiation into hepatocyte-like and cardiomyocyte-like cells. This novel state is similar to the primed state of pluripotency both at the transcriptome level and regarding the X chromosome inactivation status. The authors then show convincing data showing that the PI3K/AKT pathway regulates the pluripotent state both in hESCs and in the Epi of the blastocyst. Overall, this study establishes a novel systematic and rational way to investigate the regulation of human pluripotency and defines a novel culture condition that recapitulates that state in vitro. It also reveals that the cross-talk between the AKT and ERK pathway explains the previous misunderstanding and seemingly inappropriate use of FGFs to maintain the human pluripotent state in vitro.

We thank the reviewer for their excellent summary of our work and for appreciating the novelty of our findings. We agree that this study establishes a unique way to investigate the regulation of human pluripotency with an emphasis on applying knowledge from human epiblast establishment or maintenance. As outlined below we have included additional experiments and transcriptome analysis which further support our conclusions. We are grateful for the reviewer's helpful suggestions, which we believe have improved our manuscript.

Line 131: Please add a reference to the statement 'the interleukin 6 receptor subunit IL6R, which can bind insulin-related ligands.'

We have included the following reference on line 145 of the manuscript.

48. Abroun, S. *et al.* Receptor synergy of interleukin-6 (IL-6) and insulin-like growth factor-I that highly express IL-6 receptor α myeloma cells. *Blood* 103, 2291–2298 (2004).

49. Sprynski, A. C. *et al.* The role of IGF-1 as a major growth factor for myeloma cell lines and the prognostic relevance of the expression of its receptor. *Blood* 113, 4614–26 (2009).

Line 135 and Supplementary figure 1b:

- Provide the full list of all genes, Log2foldchange, value, padj, basemean, lfcSE, and stat for Epi enriched, TE enriched, and PE enriched in a XCell tab, to complement SF1a.

We have included this additional analysis in Supplementary Table 1. This table now includes comparisons between PE and TE and different culture conditions.

- Provide the plots for the expression levels of all ligands known to activate the MAPK pathway including but not restricted to PDGF, EGF, HB-EGF, FGF ligands in Epi, TE, PE, even if they are lowly- or non-expressed.

We agree that this data is useful for the community. We have addressed this point and points below by generating an integrated website where we integrate all of the human preimplantation embryo RNA-seq gene expression data onto all KEGG signal transduction pathways (which includes the pathways above plus all other signaling pathways annotated in KEGG). On the website, a click on each node in the KEGG pathway brings up a boxplot of the primary transcriptome data. We included highly, lowly and non-expressed genes. As the peer reviewer notes, this will be the first comprehensive analysis of putative signalling pathways that may be active in human preimplantation embryos.

We believe that our analysis and datasets will be valuable to others wishing to further refine human pluripotent stem cell media and can be used as a basis to test hypotheses about early embryonic development. In addition, the data can be used to further refine and develop novel extraembryonic stem cell cultures.

We have included an example of one of the KEGG pathways in Supplementary Figure 2 (the gene expression associated with each node in the KEGG pathway can be accessed online). We updated the manuscript starting on line 140 to include this data. We provide a comprehensive online dataset available at: https://shiny.crick.ac.uk/embryo_signalling/

- Provide insights into the regulation of downstream proteins relative to GPCR pathways that are known to regulate the MAPK pathway (e.g. but not restricted to PKA, PKC).

We have integrated this data into a searchable website where the expression patterns of any pathway in the EPI, PE and TE lineages can be viewed. We have updated the manuscript on lines 159 and in the methods section line 1444 to include this data.

Line 136: Beyond the factors secreted by the EPI, the pluripotency might be maintained by niche factors produced by the TE and PE. To challenge your finding that IGF1 is an important regulator of human pluripotency, the authors should provide the lists and plots of all secreted ligands (a.k.a. secretome) and of the corresponding receptors produced by the Epi, TE, and PE. This data would be especially relevant to assess the putative role of TGFB, WNT, and STAT pathways.

This is an excellent suggestion and we also believe this is an important consideration for our understanding of pluripotency regulation. In addition, we believe this data will be mined to look at pathways that can be modulated to improve human trophoblast stem cell conditions and to develop models of extraembryonic endoderm yolk-sac stem cells.

We initially attempted to automate this analysis using the approach suggested by the reviewer: we noted all genes that were annotated as a “secreted protein” in the Human Protein Atlas (<https://www.proteinatlas.org/humanproteome/proteinclasses>) and identified cognate receptors from the same dataset. Unexpectedly, when we looked back through the lists, we noticed that the secretome data has a number of misannotations (for example FGF2 is not listed as a ligand) and some ligands are listed as receptors and vice versa. We therefore took a different approach to avoid false positives and negatives in the list.

We therefore manually compiled a list of all ligands and receptors in the signal transduction pathways of the KEGG database (including the ones noted by the reviewer above). We then determined all ligands that interacted with receptors using the HIPPIE protein-protein interaction database (Alanis-Lobato et al., Nucleic Acid Research 2017). We believe this is an accurate way to curate ligand-receptor pairs.

To our knowledge this is the first list of its kind; we could not identify a published comprehensive annotation of ligand-receptor pairs in any cell type. We therefore believe that this annotated list will be useful for other cellular contexts.

We then incorporated the gene expression in the TE, PE and EPI for all of the ligand-receptor pairs including the absence of expression. We have included this data in Supplementary Table 2 and revised the manuscript starting on lines 142 and included a methods section detailing how we generated the list (line 1430 onwards). As noted above, we have also included a searchable graphical representation of KEGG pathway enrichment.

Line 136: Because FGFR4 is expressed in the Epi, the authors should use REACTOME or KEGG analysis to assess the regulation of classical MAPK/ERK target genes in the Epi as compared to the TE and PE. This analysis should be extended to the WNT, TGFB, and STAT pathways that were proposed to also regulate human pluripotency.

We have performed several analyses to address this point. Firstly, we used the EPI vs TE or PE and TE vs PE results from DESeq2 to perform gene set enrichment analysis (GSEA). For this, we employed the gene sets compiled by the Bader Lab (http://download.baderlab.org/EM_Genesets/current_release/Human/symbol/Pathways/, 01/08/2019 snapshot), which represent a compendium of multiple pathway databases (Reactome, Molecular Signatures Database, Institute of Bioinformatics, NetPath Signal Transduction Pathways, Panther, NCI Nature Pathway Interaction Database, WikiPathways, HumanCyc Encyclopedia of Human Genes and Metabolism) . Then, we used the EnrichmentMap app in Cytoscape to represent the relationship between enriched signalling pathways. Altogether we provide an overview of which pathways are enriched in the EPI versus TE or PE. We find that enrichment of IGF activity and Type II diabetes which includes genes associated with insulin regulation are enriched in the EPI compared to the TE.

We have included this data in Supplementary Figure 1 and updated the text starting on lines 136 to describe the findings. We also included an update to the methods section on lines 1425 to detail how we performed this analysis. A comprehensive analysis of all KEGG pathways is provided in the website mentioned above.

Line 156: Integrins and FAK have been associated with MAPK activity (e.g., DOI: 10.1038/372786a0). The authors should test whether the L511 regulates MAPK activity, for example as compared to Matrigel and other laminins previously used for hESC culture. This might in part explain the differences in ligands and signalling activity that they observed. Other controls including the addition into the culture medium of antibodies blocking or activating integrins might be insightful as well.

We agree with the reviewer that the basement membrane may impact on the signalling pathways active in hESCs, a point also noted by the second peer reviewer. To address this, we have generated additional single-cell RNA-seq data from hESCs cultured in mTeSR1 media cultured on Laminin-511. We also took this opportunity to include a biological triplicate of the RNA-seq analysis of hESCs derived directly in AI media (CH3 line).

PCA, UMAP and graph global gene expression analysis shows that the cells cultured in mTeSR1 on Matrigel are transcriptionally similar to those cultured on Laminin-511 (Figure 2a and Supplementary Figures 8a-c). The subset of the hESCs cultured in mTeSR1 on Matrigel that cluster separately in PC2 in Supplementary Figure 8 were samples sequenced in a different run compared to the other mTeSR1 samples. We include below a figure highlighting the two groups of mTeSR1 cells cultured on Matrigel sequenced in different runs for the peer reviewers. This figure highlights the importance of non-linear dimensionality reduction approaches such as UMAP and methods such as graph, which control for confounding sources of variance.

We find similar results in the comparison between hESCs cultured in AI media versus mTeSR1, irrespective of the basement membrane (Figure 2a and Supplementary Figures 8a-c). Altogether this indicates that hESCs cultured in AI media are transcriptionally indistinguishable from hESCs grown in mTeSR1 on either Matrigel or Laminin-511.

With regards to the question of functionally testing the impact of integrins and FAK on MAPK activity using blocking antibodies, we too believe that this would be an interesting experiment to test in the future. We have noted this in the discussion section from line 594.

The two RNA-seq runs containing mTeSR1 cells cultured on Matrigel are highlighted in pink.

Line 180: The authors should add controls to their FACS experiment including hESCs cultured in F2/KSR and t2iLGö.

We have included control hESCs cultured in mTeSR1 on Laminin, KSR+FGF2 on MEFs and t2iLGo. This data is now included in Supplementary Figure 6 and shows that AI cells express these markers in a similar proportion to primed hESCs. We have revised the text starting on line 247 describing these additional control flow cytometry analyses.

Line 184: The putative superiority of AI over other culture conditions for clonal derivation should be quantified via a CFU assay.

We thank the reviewer for this suggestion. We agree that determining the ability to clonally derive lines from hESCs is important and would demonstrate the utility of this culture condition to genetically modifying hESCs for knock-in and knock-out studies.

To evaluate this, we nucleofected a plasmid engineered to express the Cas9 gene, a guide RNA targeting a non-essential gene in hESCs (*ARGFX*) and a puromycin selection cassette. Following 48h of puromycin selection, we collected cells 10 days later for a T7

endonuclease assay to determine whether CRISPR-Cas9-mediated genome editing had taken place. We also performed alkaline phosphatase staining indicating that cells culture in AI medium are amenable to clonal derivation. We have included this data in Supplementary Figure 5c and 5d, and a description of these results from line 225. We also included an update to the methods section.

Line 190: The author should clearly mention in the conclusions that they used multiple lines and name them (H1, H9, CH2, Shef6).

This is an important point and we have provided clarification that we used hESC lines H1, H9, Shef6, iPSC line RCiB10 on line 213, in addition to the hESC and iPSC lines we derived *de novo*.

Line 200: The authors should mention the number of cell lines that they derived *de novo*. Was the derivation efficiency similar to FGF-driven derivation? Did the authors see reproducible differences in the transcriptome of *de novo* lines as compared to converted lines? This might point at irreversible changes due to a derivation in FGF as compared to IGF1.

We have updated the manuscript on line 250 to clarify that 3 independent hESC lines were derived in AI medium. We unfortunately cannot confidently comment on the derivation efficiency compared to conventional derivation conditions because we stopped deriving new lines in AI medium after the third hESC line was established.

DESeq2, principal component analysis, UMAP and graph analysis of global gene expression indicate that the *de novo* derived lines cluster together with and are transcriptionally similar to the converted lines (Figure 2a and Supplementary Figure 8a-c).

Line 258: Due to the sensitivity of the adjustment of batch effects, it would be important to briefly mention the method in the text.

We agree with the reviewer that this is important to clarify and have provided additional information in the methods section starting on line 1392.

Line 287: The pair-wise analysis should be also done using Panther and KEGG in order to depict the differences in signalling activity.

We thank the reviewer for this suggestion. The new version of our manuscript now includes DESeq2-based comparisons of cells cultured in AI and cells cultured in KSR, mTeSR1 on laminin, mTeSR1 on Matrigel and t2iL+Go (Supplementary Table 3). Similar to the comparison we performed for the three cell types of the human blastocyst (see above), we used the DESeq2 results to perform GSEA with the gene sets compiled by the Bader Lab at:

http://download.baderlab.org/EM_Genesets/current_release/Human/symbol/Pathways/01/08/2019_snapshot). These gene sets represent a compendium not only of KEGG and Panther pathways but also data from Reactome, the Molecular Signatures Database, the

Institute of Bioinformatics, NetPath Signal Transduction Pathways, the NCI Nature Pathway Interaction Database, WikiPathways and HumanCyc.

We have included the gene set enrichment analysis in Supplementary Table 4 and updated the manuscript starting on line 340.

Line 293-298: The lower WNT activity in Epi as compared to AI- and TeSR-cultured hESCs is very interesting as it contradicts the current paradigm that WNT should be activated in naive hESCs. The enrichment for genes associated with IL6 activity in both Epi and AI-hESCs is very interesting as well. It is thus tempting to ask the authors to perform longer-term culture of hESCs in AI supplemented with WNT inhibitors (e.g., IWP2, XAV939), PD17/PD03 (in relation to line 371 and 382, and possibly at concentration 0.5 μ M for PD03, in light of doi:10.1038/s41592-018-0104-1) and in the presence of IL6.

We agree with the reviewer that the comparison of the AI and mTeSR cells to EPI to identify pathways that can be modulated to promote naïve pluripotency using an informed strategy based on the biology of the embryo would be interesting. Indeed, this is precisely the future direction of this project. We would first like to perform functional studies in human embryos to determine if modulating these pathways has an impact on epiblast development. This would indicate whether the signalling pathways are at all active, which they may not be given that the differential gene expression analysis is solely based on comparative transcriptome analysis. If we confirm activity of the pathways, and modulation of the pathways shows a positive effect on the establishment or maintenance of the pluripotent epiblast, then we would seek to recapitulate similar signaling activity in *in vitro* cultured hESCs.

We have preliminary data suggesting that nuclear β -catenin expression is restricted to the trophectoderm and primitive endoderm in human embryos, further supporting the hypothesis that blocking WNT signalling may promote the EPI at the expense of the trophectoderm. Consequently, we intend to treat human embryos with IWP2, ChIR or exogenous WNTs. We will be interested to understand if this signalling modulation leads to a cell fate switch. We are also interested in the enrichment of IL6 and other components of the STAT signalling pathway. Again, we would like to assess if modulating this pathway has an impact on human embryo epiblast development. It will take us some time to understand what role, if any, these signalling pathways have in the development of the embryonic epiblast and then to subsequently optimise hESC derivation and maintenance conditions based on these findings. We would like to pursue this line of enquiry in future studies and we anticipate that others will use our dataset to inform their approaches to refine hESC media and to establish optimised models of human extraembryonic stem cells.

Line 351: The authors should check whether the elements of the mTOR pathway are differentially regulated in the Epi as compared to the TE and PE.

The reviewer makes a good point, and we have provided this information in the searchable online database. The KEGG diagram corresponding to the mTOR signalling pathways shows that its components are expressed in the EPI. Clicking on each node of

the pathway shows the specific genes enriched in the EPI, TE and/or PE. This is consistent with the Western blot data in Figure 4a showing activity of this pathway in human preimplantation embryos.

Line 428: The authors should use their refined WB technique to check the levels of pERK in human blastocysts as compared to mouse blastocysts (in which the pathway activity is well established) and to hESCs in AI-, KSR-, and t2iLGö-culture conditions. The hESCs samples can be dosed from 1 to 100 cells for example, in order to give an idea of the respective levels.

We thank the reviewer for this suggestion, and agree that evaluating the expression of pERK in human embryos is important. We detected pERK expression in human embryos and provide this data in Figure 4a and Supplementary Figure 11. We have updated the text starting on line 491 accordingly. We acknowledge that dosing hESCs to provide an idea of respective levels is a good idea, but given the principal focus of this study on the role of IGF1 in replacing the function of FGF2 to promote hESC self-renewal, we feel that this experiment is beyond the scope of our current study. This is an experiment we would like to optimise in the future.

As a summary, in order to make this paper a landmark in hESC culture by unbiasedly assessing signalling pathways regulating human pluripotency, I would suggest that the authors:

- perform additional computational analysis of the single cell sequencing data to:

(i) provide the full list of all genes, Log2foldchange, value, padj, basemean, lfcSE, and stat for Epi enriched, TE enriched, and PE enriched,

(ii) provide the plots for the expression levels of all ligands activating the MAPK pathway including but not restricted to PDGF, EGF, HB-EGF, FGF ligands in Epi, TE, PE,

(iii) provide insights into the downstream proteins of GPCR pathways that are also known to regulate the MAPK pathway (e.g., PKA, PKC)

(iv) provide the list and plots of all secreted ligands and of the corresponding receptors produced by the Epi, TE, and PE, especially for the TGFB, WNT, and STAT pathways

(v) provide panther and KEGG analysis of the AI- as compared to F2/KSR- and t2iLGö-cultured hESCs

- perform an additional experiment to weight the induction of MAPK activity via integrins

- add controls to the FACS experiment (Fig 1c) to quantify and compare the levels of pluripotency-related genes

- test the clonal efficiency of AI- as compared to F2/KSR- and t2iLGö-cultured hESCs

- attempt the long-term culture and characterise AI-hESCs in the presence of WNT inhibitors (IWP2, XAV939), ERK inhibitors (PD17, PD03) and IL6 [modified AI culture]

- verify the levels of pERK in human blastocysts as compared to mouse blastocysts and to multiple doses of hESCs in AI-, modified AI-, F2/KSR- and t2iLGö-cultured hESCs.

Nicolas Rivron

We thank the peer reviewer for their very helpful suggestions, which have improved our manuscript. We hope they agree that we have sufficiently addressed the points they raised.

--

Reviewer #2 (Remarks to the Author):

This manuscript shows that FGF2 is dispensable in the culture of human embryonic stem cells (hESCs) when replaced with IGF1 and insulin. This conclusion is based on a screen from prior single cell RNA seq data of a day 6 embryo and based on their work culturing hESCs in IGF1-containing media. I think their results are interesting and will be valuable to the field.

We thank the reviewer for their positive comments.

My main concern, however, is that this result is not very novel. They show that cells in AI medium have a highly similar transcriptomics profile to hESCs conventionally grown in mTesr (with the exception of morphology). At the authors' own admission, "IGF1 and FGF2 may activate overlapping downstream mechanisms for the maintenance of hESCs". Cells cultured in these two media also respond similarly to differentiation cues. Based on that, it is unlikely that the altered culture conditions will change our current hESC media practices and the value of the paper is in linking the IGF pathway to pluripotency.

However, this point is not novel either. The authors missed previous references that have already shown that PI3K/AKT signaling is active in hESCs, that it is necessary for pluripotency maintenance and that this signaling is activated by insulin/IGF (McLean et al. 2007; Singh et al. 2012). Furthermore, FGF-free media were also developed and used to maintain pluripotency. It included heregulin in addition to IGF and Activin (Singh et al. 2012). Cross-talk between PI3K and MAPK pathways in hESCs has also been previously demonstrated at the level of pAKT and pERK (Singh et al. 2012). The authors should discuss and cite this prior work. Also given the findings of these papers, I think all instances of the word "first" should be removed from the current paper.

Using the human embryo to optimize the culture conditions for hESCs is a novel approach with a strong rationale. However, the fact that AI culture conditions maintain pluripotency is not surprising given the above-mentioned work. The main new insight that the authors add is to demonstrate that the FGF receptor can be inhibited without consequences to pluripotency, thus providing clear evidence that FGF signaling is dispensable for hESC maintenance and that the main function of FGF in conventional media is to activate PI3K signaling

We appreciate the reviewer's comments and thank them for their helpful suggestions regarding relevant literature. We have revised the discussion to state that:

"Notably, previous studies showed that inhibition of PI3K is required to differentiate hESCs to definitive endoderm and that the presence of insulin in KSR and IGF in FBS promote self-renewal and antagonize differentiation (McLean et al., 2007). However, these hESC culture media contained exogenous FGF2, which our data suggests is not required. It would be interesting to determine in the future whether there may be differences in the directed differentiation efficiencies of hESCs cultured in AI versus conventional conditions (containing FGFs). We predict that directed differentiation may be more efficient when starting with AI media given the lack of redundant pathways to activate PI3K (i.e. FGF+insulin/IGF versus insulin/IGF/insulin) and this will be interesting to test in the future."

We also include the following point in the discussion:

"It has been previously shown that IGF1 and heregulin can reproduce the same positive effect on long-term self-renewal of hESC pluripotency as high levels of FGF2 (Singh et al., 2012). Notably, our conditions lack heregulin, which is a member of the epidermal growth factor (EGF) family and thereby can activate the RTKs, similarly to FGFs. Intriguingly, the authors suggest that hESCs have endogenous autocrine WNT signaling and that DKK1 (an inhibitor of WNT signaling) further promoted hESC self-renewal (Singh et al., 2012). Inhibiting WNT signaling promotes naïve to primed hESC transition (Xu et al., 2016). This, together with our differential gene expression analysis indicating WNT signaling differences in the embryonic epiblast compared to hESCs in AI media, suggests that modulating WNT may further promote self-renewal."

Despite the findings of the papers quoted above, and additional supporting evidence dotted in the literature, the field is arguably still not convinced that FGF is not required and continue to use it in primed pluripotency media. Here, as the reviewer notes, we show definitive proof that FGF is not required in the absence of any upstream activators of RTK and expand on this to show the relatedness to the developing EPI. Also, we note the activity of the AKT/mTOR pathway in naïve hESCs, which we do not believe has been demonstrated previously.

We agree that the AI media conditions we developed have been hinted at in the literature, although we would argue that they have yet to be explored as comprehensively as in our study. As suggested by Reviewer 1, we have expanded considerably on the section related to signalling pathways in the developing embryo and believe this will be a valuable novel resource for the developmental biology and stem cell research communities. With our work with IGF, we also identify a further relationship between the signalling pathways required for hESC culture and those occurring in the niche environment of the developing pluripotent epiblast, which though expected has often yet to be convincingly demonstrated. As the reviewer notes, this is a novel approach with a strong rationale, and we believe our conclusions may lead to changes in how primed hESCs are cultured in the future and a switch from FGF-containing media to media where this is absent. Furthermore, future adoption of this rational embryo-driven approach may also lead to the development of further optimised conditions both in hESC culture, and for human extraembryonic stem cell derivation.

Additionally, the authors should consider comparing transcriptome data for AI conditions to other media conditions in which cells are also grown on laminin-511, so as to highlight any differences between using FGF and IGF rather than differences resulting from the extra-cellular matrix. This would strengthen the paper and the impact of its findings. If this comparison was already made then the authors should discuss it in more detail (e.g. AI vs mTeSR1 on Laminin-511).

We agree with the reviewer that this is an important comparison. To address this, we have generated additional single-cell RNA-seq data from hESCs cultured in mTeSR1 media cultured on Laminin-511. We also took this opportunity to include a biological triplicate of the RNA-seq analysis of hESCs derived directly in AI media (CH3 line). We performed transcriptome analysis of hESCs cultured in mTeSR1 media on Laminin-511 and have revised Figure 2 and Supplementary Figure 8 and 9 to include this additional data.

PCA, UMAP and graph global gene expression analysis shows that the cells cultured in mTeSR1 on Matrigel are transcriptionally similar to those cultured on Laminin-511 (Figure 2a and Supplementary Figures 8a-c). The subset of the hESCs cultured in mTeSR1 on Matrigel that cluster separately in PC2 in Supplementary Figure 8 were samples sequenced in a different run compared to the other mTeSR1 samples. We include below a figure highlighting the two groups of mTeSR1 cells cultured on Matrigel sequenced in different runs for the peer reviewers. This figure highlights the importance of non-linear dimensionality reduction approaches such as UMAP and methods such as graph, which control for confounding sources of variance. The two RNA-seq runs containing mTeSR1 cells cultured on Matrigel are highlighted in pink.

We find similar results in the comparison between hESCs cultured in AI media versus mTeSR1, irrespective of the basement membrane (Figure 2a and Supplementary Figures 8a-c). Altogether this indicates that hESCs cultured in AI media are transcriptionally indistinguishable from hESCs grown in mTeSR1 on either Matrigel or Laminin-511.

It is crucial to discuss these points and revise the paper accordingly before further consideration. In all, while I think the work is solid, my concern is that not very much about the work (culture conditions or the mechanism itself) is very novel and it is up to the editor to decide whether it is suitable for Nature Communications.

Missed references:

McLean, Amanda B, Kevin A D'Amour, Karen L Jones, Malini Krishnamoorthy, Michael J Kulik, David M Reynolds, Alan M Sheppard, et al. 2007. "Activin a Efficiently Specifies Definitive Endoderm From Human Embryonic Stem Cells Only When Phosphatidylinositol 3-Kinase Signaling Is Suppressed." 25 (1). John Wiley & Sons, Ltd.: 29–38. doi:10.1634/stemcells.2006-0219.

Singh, Amar M, David Reynolds, Timothy Cliff, Satoshi Ohtsuka, Alexa L Mattheyses, Yuhua Sun, Laura Menendez, Michael Kulik, and Stephen Dalton. 2012. "Signaling Network Crosstalk in Human Pluripotent Cells: a Smad2/3-Regulated Switch That Controls the Balance Between Self-Renewal and Differentiation.." Cell Stem Cell 10 (3): 312–26. doi:10.1016/j.stem.2012.01.014.

We are very grateful to the reviewer for pointing out these missed references which we agree are important to include. As noted above they have been address in the revised manuscript.

Reviewers' Comments:

Reviewer #1:

Remarks to the Author:

Dear Authors, dear editor,

The authors have addressed the most important comments, and clarified and substantiated their claims. The detailed single cell transcriptomic analysis is a valuable source of information for the community and clearly demonstrate the power of using a rational approach to define the conditions necessary for the in vitro maintenance of the pluripotent states. The website is of great interest, and complementary to other websites that are being prepared at the moment. However, due to possible problems of maintenance of websites, it would be important to ensure that all the data necessary to substantiate the author's claims are also present in the paper itself, including the raw (even if previously published) and the processed data. I would like to propose the following minor textual changes.

. The signalling pathway activity being very dynamical, it would be important to mention, in the text and in the figures, the embryonic stages that were investigated in the transcriptomic analysis, to the best of the authors knowledge.

. It is difficult to firmly assess signalling pathway activity based on transcriptomic data. It might not be reasonable to discard the possibility that MAPK/ERK plays a role in the Epi. I believe that, in order to insure the durability of the author's claims in the future, some sentences should be rewritten. For example:

1. the statement on lines 166-168 "although FGFR signalling was highlighted as enriched in the EPI compared to the PE, the enrichment map category was related to the negative regulation of this signaling pathway (Supplementary Table 1)" should be modified as the negative regulation of FGFR3 and FGFR4 signalling observed in Epi vs. PE can reflect the classical negative feedback loops reflecting a pathway activation.

2. The statement on line 168 should be tempered. "Consistent with this, we did not find FGF-related receptors specifically highly enriched in the EPI. Boxplot analysis showed FGFR1, FGFR2 and FGFR4 were more highly expressed in the PE, while transcripts for all FGF ligands, with the exception of FGF18, were not expressed above the required 5 RPKM threshold (Supplementary Fig. 3; https://shiny.crick.ac.uk/embryo_signalling/). The expression level of FGFR1 (~25 RPKM) are lower in Epi as compared to PE but probably sufficient to support MAPK/ERK activity. Note that these receptors might however not mediate signalling activity at this precise stage, but rather prepare the Epi cells for further developmental transitions. The lack of FGF ligands at this specific stage as presented in Sup figure 3 along with the functional inhibition using small molecules (ref 22-23) are the most convincing arguments supporting the lack of FGF-mediated activity. However, other FGF ligands are expressed in stages preceding the blastocyst stage and are thus likely to play a role (e.g., FGF8, FGF12, FGF13). Finally, the uterine environment might contain FGFs that play a role in the Epi. In the future, a timed analysis from morula to late blastocyst stages would be necessary to further strengthen these claims.

. Please mention the downstream MAPK pathway components that are used to support the claim on line 172.

. I find the "E2" in figure 4e unclear. It could be replaced by "Treatment from E2 to E6.5"

. Line 531: "Altogether, this suggests that rather than operating via an alternative pathway in the human embryo, FGF is unlikely to be required for EPI development". Here, I would modify to "Altogether, this suggests that rather than operating via an alternative pathway in the human embryo, FGF is unlikely to be crucial for EPI development".

Reviewer #2:

Remarks to the Author:

After revision, the response has addressed everything asked to be addressed, and I can happily give my recommendation for publication.

We thank both reviewers for all of their helpful comment and suggestions. We have included a response in red to the additional comments below.

REVIEWERS' COMMENTS:

Reviewer #1 (Remarks to the Author):

Dear Authors, dear editor,

The authors have addressed the most important comments, and clarified and substantiated their claims. The detailed single cell transcriptomic analysis is a valuable source of information for the community and clearly demonstrates the power of using a rational approach to define the conditions necessary for the in vitro maintenance of the pluripotent states. The website is of great interest, and complementary to other websites that are being prepared at the moment. However, due to possible problems of maintenance of websites, it would be important to ensure that all the data necessary to substantiate the author's claims are also present in the paper itself, including the raw (even if previously published) and the processed data. I would like to propose the following minor textual changes.

We thank the peer reviewer for their positive comments and for finding the website of great interest. We agree that this will be a valuable resource for the community and anticipate that it will lead to additional discoveries. We have confirmed with the Francis Crick Institute that the ShinyApp will continue to be supported and hosted. Moreover, we have included the processed data in Supplementary Data 6 and will make available all of the raw data through GEO. We are committed to sharing any data with other investigators now and in the future.

The signalling pathway activity being very dynamical, it would be important to mention, in the text and in the figures, the embryonic stages that were investigated in the transcriptomic analysis, to the best of the authors knowledge.

We state in the opener to the Results section (line 110) that the transcriptomics analysis was carried out in the blastocyst. To further clarify the stage, we have now amended the figure legends for Supplementary Figures 1 and 2 (lines 1278, 1305) to refer to the human blastocyst, rather than embryo.

It is difficult to firmly assess signalling pathway activity based on transcriptomic data. It might not be reasonable to discard the possibility that MAPK/ERK plays a role in the Epi. I believe that, in order to insure the durability of the author's claims in the future, some sentences should be rewritten. For example:

1. the statement on lines 166-168 "although FGFR signalling was highlighted as enriched in the EPI compared to the PE, the enrichment map category was related to the negative regulation of this signaling pathway (Supplementary Table 1)" should be modified as the negative regulation of FGFR3 and FGFR4 signalling observed in Epi vs. PE can reflect the classical negative feedback loops reflecting a pathway activation.

We appreciate the reviewer's point and have amended this statement to say on line 144: "Enrichment analysis showed that genes related to negative regulation of FGFR signalling were present in the EPI, which could reflect negative feedback loops indicating pathway activation (**Supplementary Data 1**). However, we did not find FGF-related receptors specifically highly enriched in the EPI."

2. The statement on line 168 should be tempered. "Consistent with this, we did not find FGF-related receptors specifically highly enriched in the EPI. Boxplot analysis showed FGFR1, FGFR2 and FGFR4 were more highly expressed in the PE, while transcripts for all FGF ligands, with the exception of FGF18, were not expressed above the required 5 RPKM threshold (Supplementary Fig. 3; https://shiny.crick.ac.uk/embryo_signalling/). The expression level of FGFR1 (~25 RPKM) are lower in Epi as compared to PE but probably sufficient to support MAPK/ERK activity. Note that these receptors might however not mediate signalling activity at this precise stage, but rather prepare the Epi cells for further developmental transitions. The lack of FGF ligands at this specific stage as presented in Sup figure 3 along with the functional inhibition using small molecules (ref 22-23) are the most convincing arguments supporting the lack of FGF-mediated activity. However, other FGF ligands are expressed in stages preceding the blastocyst stage and are thus likely to play a role (e.g., FGF8, FGF12, FGF13). Finally, the uterine environment might contain FGFs that play a role in the Epi. In the future, a timed analysis from morula to late blastocyst stages would be necessary to further strengthen these claims.

It is indeed possible that the uterine environment may contain FGFs. However, we would posit that as epiblast specification occurs successfully *in vitro* for example in Global Media culture medium (which does not contain FGF), and the resulting blastocysts can be implanted to generate successful pregnancies in IVF clinics. It therefore seems unlikely that FGF is absolutely required at this point, or else fertilised zygotes would not develop *in vitro*.

We find in this study that embryos treated with FGF from the 4-cell stage lose expression of NANOG (Fig 4), suggesting that this may in fact be a detrimental stimulus. It is possible that this is an FGF2-specific effect, and we acknowledge the reviewer's suggestion that other FGF ligands present may play a role. However, previous studies treating embryos with the FGF-receptor inhibitor PD173074 from the 6- to 8-cell stage until the blastocyst stage (Roode 2012) did not observe a detrimental effect on epiblast development, again suggesting that FGF signalling is not a crucial modulator during pre-implantation development. It may be that FGF is required for later post-implantation epiblast fates, but investigating this is beyond the scope of our study, though new techniques for embryo implantation or 3D-embryo modelling may provide insights. We would prefer to leave our current conclusions as they stand, but remain open to further analysis from the field in this regard.

Please mention the downstream MAPK pathway components that are used to support the claim on line 172.

We have mentioned the examples of *KRAS* and *MAPK1* as components downstream of MAPK signalling.

I find the “E2” in figure 4e unclear. It could be replaced by “Treatment from E2 to E6.5”.

We thank the reviewer for their suggestion and have amended Figure 4 to include the treatment duration. We have also changed embryonic day to days post fertilisation in order to distinguish embryonic staging for embryos culture *in vitro*.

Line 531: “Altogether, this suggests that rather than operating via an alternative pathway in the human embryo, FGF is unlikely to be required for EPI development”. Here, I would modify to “Altogether, this suggests that rather than operating via an alternative pathway in the human embryo, FGF is unlikely to be crucial for EPI development”.

We have revised the sentence on line 474 to address the very good point raised by the reviewer.

In closing, we thank the reviewer again for their comments and suggestions, which have improved our manuscript.

Nicolas Rivron

--

Reviewer #2 (Remarks to the Author):

After revision, the response has addressed everything asked to be addressed, and I can happily give my recommendation for publication.

We thank the reviewer for their recommendation.